# Distinct functions of cardiac β-adrenergic receptors in the T-tubule vs. outer surface membrane

**George WP Madders[1†], Marion Barthé[1†], Flora Lefebvre[2], Emilie Langlois[2], Florence Lefebvre[1], Patrick Lechêne[1], Maya Dia[1], Xavier Iturrioz[3,4], Catherine Llorens-Cortes[3,4], Tâp Ha-Duong[5], Laurence Moine[2], Nicolas Tsapis[2], Rodolphe Fischmeister[1]\***

[1]Université Paris-Saclay, Inserm, UMR-S 1180, Orsay, France; [2]Université Paris-Saclay, CNRS, Institut Galien Paris-Saclay, Orsay, France; [3]Collège de France, CIRB, INSERM U1050/CNRS UMR 7241, Paris, France; [4]Université Paris Saclay, CEA, Département Médicaments et Technologies pour la Santé, SIMoS, Gif-sur-Yvette, France; [5]Université Paris-Saclay, CNRS, BioCIS, Orsay, France

**Abstract** β-Adrenoceptors (β-ARs) regulate cardiac function during sympathetic nerve stimulation. β-ARs are present in both the cardiac T-tubule (TTM) and outer surface membrane (OSM), but how their location impacts their function is unknown. Here, we developed a technology based on size exclusion to explore the function of β-ARs located in the OSM. We synthesized a PEG-Iso molecule by covalently linking isoprenaline (Iso) to a 5000 Da PolyEthylene-Glycol (PEG) chain to increase the size of the β-AR agonist and prevent it from accessing the T-tubule network. The affinity of PEG-Iso and Iso on $\beta_1$- and $\beta_2$-ARs was measured using radioligand binding. Molecular dynamics simulation was used to assess PEG-Iso conformation and visualize the accessibility of the Iso moiety to water. Using confocal microscopy, we show that PEGylation constrains molecules outside the T-tubule network of adult rat ventricular myocytes (ARVMs) due to the presence of the extracellular glycocalyx. β-AR activation in OSM with PEG-Iso produced a lower stimulation of $[cAMP]_i$ than Iso but a larger stimulation of cytosolic PKA at equivalent levels of $[cAMP]_i$ and similar effects on excitation–contraction coupling parameters. However, PEG-Iso produced a much lower stimulation of nuclear cAMP and PKA than Iso. Thus, OSM β-ARs in ARVMs control mainly cytosolic cAMP/PKA pathway and contractility, while TTM β-ARs control mainly nuclear cAMP, PKA, and consequent nuclear protein phosphorylation. Size exclusion strategy using ligand PEGylation provides a unique approach to evaluate the respective contribution of T-tubule vs. OSM proteins in cardiac cells.

**\*For correspondence:**
rodolphe.fischmeister@inserm.fr

†These authors contributed equally to this work

**Competing interest:** The authors declare that no competing interests exist.

## Editor's evaluation

This important study describes a novel approach using PEGylated isoprenaline to selectively activate β-adrenergic receptors in the surface sarcolemma relative to T-tubule sarcolemma of ventricular myocytes. Overall, the strength of evidence presented is convincing, and the authors present an interesting and impactful study that will be of interest to cardiac cell biologists and pharmacologists.

## Introduction

The β-adrenergic receptor (β-AR) is a key player in the regulation of cardiac function during sympathetic nerve stimulation. The classical pathway for β-AR receptor signaling is activation of adenylyl cyclases (AC) via $G_{\alpha s}$, resulting in increased intracellular cAMP levels ($[cAMP]_i$) (*Leroy et al., 2018*).

Three types of β-ARs are expressed in the heart, respectively, $\beta_1$-, $\beta_2$-, and $\beta_3$-ARs. $\beta_1$- and $\beta_2$-ARs are both positively coupled to the AC/cAMP cascade and involved in cardiac performance (*Xiang, 2011*) while $\beta_3$-ARs may act via either cAMP or cGMP depending on the cell type (atrial vs. ventricular myocytes) (*Schobesberger et al., 2020*; *Skeberdis et al., 2008*). The primary target of cAMP is the cAMP-dependent protein kinase (PKA) that in turn phosphorylates several key proteins involved in the excitation–contraction (EC) coupling, such as the L-type $Ca^{2+}$ channel (LTCC or $Ca_V 1.2$), phospholamban, and troponin I (*Bers, 2002*). The phosphorylation of $Ca_v 1.2$ or its regulatory protein Rad (*Leroy, 2020*; *Liu et al., 2020*) leads to enhanced LTCC current ($I_{Ca,L}$) and sarcoplasmic reticulum (SR) $Ca^{2+}$ release via the ryanodine type 2 receptor (RyR2), contributing to enhanced $Ca^{2+}$ transients and contraction (*Bers, 2008*; *Eisner et al., 2017*). Phosphorylation of phospholamban increases $Ca^{2+}$ uptake into the SR, which accelerates $Ca^{2+}$ transient decay and, together with troponin I phosphorylation, speeds up relaxation. Whereas short-term stimulation of β-AR/cAMP is beneficial for the heart, chronic activation of this pathway results in altered $Ca^{2+}$ signaling, cardiac hypertrophy, and fibrosis, leading to ventricular dysfunction (*Osadchii, 2007*) and cardiac arrhythmias (*Boluyt et al., 1994*; *Morisco et al., 2001*; *Engelhardt et al., 1999*; *Antos et al., 2001*; *Karam et al., 2020*).

The cell membrane of cardiomyocytes is characterized by invaginations of the surface membrane, occurring primarily perpendicular to myocyte longitudinal edges, at intervals of ~1.8–2 μm, that form a complex interconnected tubular network penetrating deep into the cell interior (*Hong and Shaw, 2017*). This network is called the transverse (T-) tubules (TT) system, although the invaginations may often bifurcate in the axial direction or form branches (*Hong and Shaw, 2017*). These tubular structures are found mostly in adult ventricular myocytes, where they represent about 30% of the total cell membrane (*Brette et al., 2006*; *Bourcier et al., 2019*), and occur near the sarcomeric z-discs (*Caldwell et al., 2014*), forming functional junctions with the SR called dyads. They are critical for EC coupling by concentrating LTCCs and positioning them at close proximity of RyR2 clusters at the junction of SR to form $Ca^{2+}$ release units (*Guo et al., 2013*; *Sipido and Cheng, 2013*). During an action potential, TTs propagate the cell-membrane depolarization inside the cell, allowing $Ca^{2+}$ entry to trigger successive $Ca^{2+}$ release, thus promoting synchronicity as well as ensuring efficiency of the $Ca^{2+}$-induced $Ca^{2+}$ release (CICR) phenomenon (*Fu et al., 2016*). Their enrichment with ion channels and the presence of signaling pathways components such as β-ARs (*Nikolaev et al., 2010*) and ACs (*Timofeyev et al., 2013*) make these structures essential for cardiomyocyte function and regulation.

Biochemical assays following membrane fractionation have provided indirect evidence that membrane proteins may have different properties whether they are located in TTM or outer surface membrane (OSM). Other indirect evidence came from experiments in cardiomyocytes in which the TTM was uncoupled from OSM (detubulation) using a hyperosmotic shock with molar concentrations of formamide (*Brette et al., 2006*; *Moench et al., 2013*; *Cros and Brette, 2013*; *Brette et al., 2005*; *Fowler et al., 2004*; *Brette et al., 2004*; *Thomas et al., 2003*; *Despa et al., 2003*; *Brette et al., 2002*; *Kawai et al., 1999*). However, none of these approaches allowed us to investigate in an intact cardiomyocyte whether the function of a given membrane receptor differs if the receptor is located in OSM or TTM. To do so requires being able to separately activate or inhibit the receptor in OSM or TTM, which has not been feasible so far. Here, we tackled this challenge by developing a size exclusion strategy using the PolyEthylene-Glycol (PEG)ylation technology. By covalently linking isoprenaline (Iso) to a PEG chain, we increased the size of the β-AR agonist to prevent it from accessing the TT network. Our working hypothesis is that PEGylated isoprenaline (PEG-Iso) activates only β-ARs present in OSM while free Iso activates β-ARs present in both OSM and TTM. We thus characterized the pharmacological properties of PEG-Iso and compared its functional effects in adult rat ventricular myocytes (ARVMs) with those of Iso.

## Results

### PEGylation constrains molecules outside the TT network

The rationale for using ligand PEGylation as a tool to impede ligand diffusion into T-tubules came from the observation that fluorescent $PEG_{5000}$ molecules were unable to access TTs. A typical representative experiment (out of 15) using confocal microscopy is shown in *Figure 1*. When ARVMs were exposed during 15 min to 100 μM of $PEG_{5000}$ molecules functionalized with Fluorescein Isothiocyanate (FITC), no fluorescence was seen throughout the cell (*Figure 1A, B*). On the contrary, when the cells

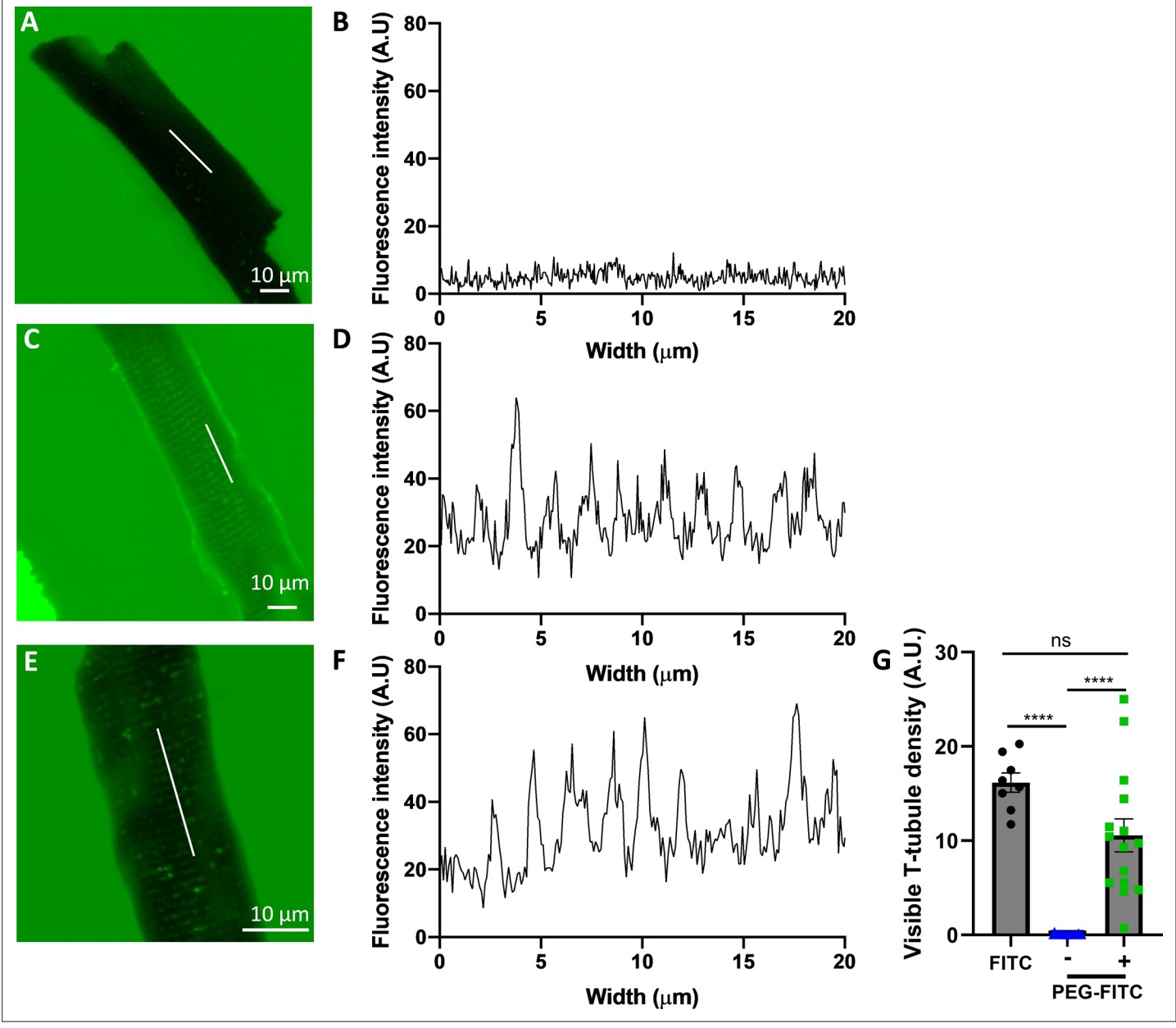

**Figure 1.** Localization of PEG-FITC in freshly isolated adult rat ventricular cardiomyocytes. Typical confocal images (**A, C, E**) and plot profiles representing the fluorescence intensity measured across the cell (**B, D, F**) of adult rat ventricular myocytes (ARVMs) incubated during 15 min with either 100 µM PEG$_{5000}$-FITC (**A, B**), 100 µM free fluorescein (**C, D**), or 100 µM PEG$_{5000}$-FITC after a 1-hr treatment with 0.25 U/ml neuraminidase (+nmd) (**E, F**). Summary fluorescence intensity for cells stained with FITC and PEG-FITC either in control (−, blue) conditions or after treatment with neuraminidase (+, green) (**G**). Bars represent the mean ± SEM. $n/N$ = 8–18/2–4. Kruskal–Wallis test with multiple comparisons was used: ****$p < 0.0001$; ns, non-significant.

The online version of this article includes the following figure supplement(s) for figure 1:

**Figure supplement 1.** PEG-Isoprenaline synthesis.

were exposed for 15 min to 100 µM fluorescein, fluorescence was seen with clear staining of the TTs (***Figure 1C, D***). We suspected that the lack of access to PEG-FITC in the TTs was due to steric hindrance from the presence of glycocalyx matrix. To test this hypothesis, ARVMs were first exposed during 1 hr to a solution containing 0.25 U/ml neuraminidase, an enzyme that degrades sialic acid in the glycocalyx matrix (***Parfenov et al., 2006***). As shown in ***Figure 1E, F***, this allowed PEG-FITC to enter the TTs as evidenced by the striated profile now observed. Summary data (***Figure 1G***) reflects

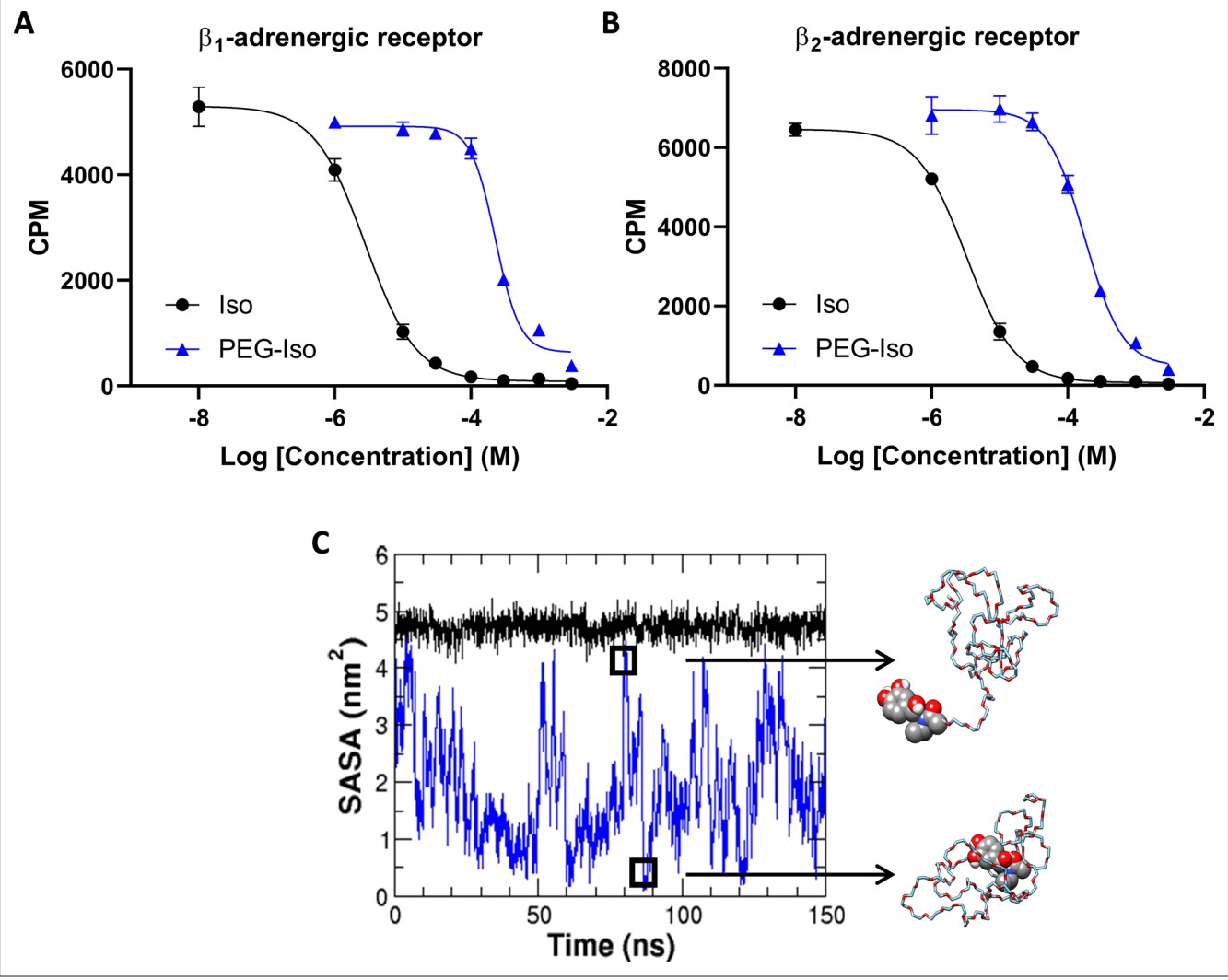

**Figure 2.** Binding properties of Iso and PEG-Iso on β-ARs. β₁- or β-AR Chinese Hamster Ovary (CHO) membrane preparations (2.5 µg/triplicate) were incubated for 2 hr with 0.25 nM [$^{125}$Iodo]cyanopindolol and increasing concentrations of unlabeled Iso or PEG-Iso. (**A**) Example of a typical competition binding of Iso and PEG-Iso on β₁-ARs. (**B**) Example of a typical competition binding of Iso and PEG-Iso on β₂-ARs. $K_i$ values were, respectively, 2.0 ± 0.7 µM and 1.4 ± 0.7 mM for Iso and PEG-Iso on β₁-ARs; 2.1 ± 0.4 µM and 0.4 ± 1.0 mM for Iso and PEG-Iso on β₂-ARs (*N* = 5). (**C**) Time evolution of solvent-accessible surface area (SASA) of Iso (black trace) and PEG-Iso (blue trace) using molecular dynamics simulations. Two representative conformations of PEG-Iso with large or small SASA are shown.

these observations where fluorescence intensity is reduced from clearly stained TTs in FITC to almost 0 fluorescence in PEG-FITC. Degradation of the glycocalyx by neuraminidase restores fluorescence intensity to the same level as that seen in free FITC.

## PEG-Isoprenaline binds to β₁- and β₂-adrenergic receptors

The above experiments demonstrate that PEG$_{5000}$ molecules are unable to access the TTs in our experimental conditions. We thus synthesized PEGylated isoprenaline (PEG-Iso) by covalently linking isoprenaline to PEG$_{5000}$ molecules (*Figure 1—figure supplement 1*). Our rationale was that since PEG-Iso does not access TTs, it will not reach β-ARs located in the TTM. But the question remained whether PEG-Iso would be able to bind to β-ARs located in the OSM. To address this question, we performed radioligand binding studies in purified membranes from Chinese Hamster Ovary (CHO) cells overexpressing either human β₁- or β₂-ARs. Competition curves between [$^{125}$Iodo]cyanopindolol and Iso or PEG-Iso were used to measure $K_i$ values of both ligands for the β₁- and β₂-ARs. As shown in *Figure 2A, B*, PEG-Iso binds to both β₁- and β₂-ARs, but with an affinity that is ~2 orders of magnitude lower than

Iso. Functional experiments using homogeneous time-resolved fluorescence (HTRF) technology on HEK 293 cells that stably expressed either the $\beta_1$- or $\beta_2$-ARs showed that the $EC_{50}$ for cAMP stimulation induced by Iso was 7.4 ± 3.6 pM vs. 39.4 ± 24.2 nM ($n = 4$) for PEG-Iso in $\beta_1$-AR expressing cells and 3.4 ± 7.7 pM vs. 20.2 ± 69.8 nM ($n = 4$) for PEG-Iso in $\beta_2$-AR expressing cells. Thus, the ability of PEG-Iso to activate $\beta_1$- and $\beta_2$-ARs was reduced by the same order of magnitude for both receptors, which should not bias the following cellular experiments.

We hypothesized that the decreased affinity of PEG-Iso could be due to wrapping of the PEG chain around the Iso moiety reducing its exposure to solvent (water). To test this, we used molecular dynamics simulations to evaluate the solvent-accessible surface area (SASA) of Iso moiety in PEG-Iso compared to free Iso. *Figure 2C* shows that free Iso SASA remains steady over the whole studied time range (black line). In contrast, the Iso moiety in PEG-Iso (blue line) has an SASA that largely fluctuates and on multiple instances reaches values close to zero, indicating its frequent shielding by the PEG chain. We calculated that only 1.2% of PEG-Iso conformations have an Iso SASA larger than 90% of the free Iso average value. This means that over a time lapse of 100 s, the Iso moiety is only unwrapped from the PEG chain and able to bind to a receptor for ≈1 s of the experiment.

## Comparison of the effects of PEG-Iso and Iso on cytosolic cAMP in ARVMs

The next series of experiments was designed to test whether PEG-Iso was able to produce a functional $\beta$-AR response in ARVMs. First, cytosolic cAMP ($[cAMP]_i$) was monitored in isolated ARVMs expressing the FRET-based sensor Epac-S$^{H187}$ (*Klarenbeek et al., 2015*). As seen in *Figure 3*, both Iso (*Figure 3A, B*) and PEG-Iso (*Figure 3C, D*) produced a concentration-dependent increase in $[cAMP]_i$. However, as highlighted in *Figure 3E*, there were two major differences: (1) the concentration-response curve to PEG-Iso was shifted ~100-fold toward larger concentrations as compared to Iso; (2) the maximal efficacy of PEG-Iso was significantly lower by ~30% than that of Iso. The former was anticipated based on the lower binding affinity of PEG-Iso to $\beta$-ARs shown above. The latter can be explained by the fact that PEG-Iso only activates $\beta$-ARs in OSM while Iso activates $\beta$-ARs in both OSM and TTM, thus producing a larger $[cAMP]_i$ elevation.

## Comparison of the effects of PEG-Iso and Iso on $I_{Ca,L}$

Next, the ability of Iso and PEG-Iso to stimulate the L-type $Ca^{2+}$ current, $I_{Ca,L}$ was compared at a single concentration of each, 10 nM Iso and 1 µM PEG-Iso, shown above to produce an equivalent response on $[cAMP]_i$ (*Figure 3*). As shown in the individual experiments in *Figure 4A, B*, both Iso and PEG-Iso produced a similar increase in the amplitude of $I_{Ca,L}$ (*Figure 4C*).

## Comparison of the effects of PEG-Iso and Iso on EC coupling

To investigate the impact of stimulating TTM and OSM $\beta$-ARs on EC coupling, $Ca^{2+}$ transients and sarcomere shortening measurements were simultaneously recorded in Fura-2-loaded ARVMs and their response to PEG-Iso and Iso evaluated. Here, two concentrations of PEG-Iso were used, 100 and 300 nM, and compared to two affinity-adjusted concentrations of Iso that produce similar elevations of $[cAMP]_i$, 1 and 3 nM. As shown in *Figures 5 and 6*, 100 and 300 nM PEG-Iso increased contractility (*Figure 5B*), $Ca^{2+}$ transients (*Figure 6B*), and accelerated their relaxation kinetics (*Figures 5D and 6D*). These effects were not different from those produced by 1 and 3 nM Iso, respectively, for either sarcomere shortening (*Figure 5A, C*), $Ca^{2+}$ transient amplitude (*Figure 6A, C*) or relaxation kinetics (*Figures 5C, D and 6C, D*).

## Comparison of the effects of PEG-Iso and Iso on cytosolic and nuclear PKA activity

The observation that equipotent concentrations of PEG-Iso and Iso on $[cAMP]_i$ produced equivalent effects on $I_{Ca,L}$ and EC coupling may suggest that OSM $\beta$-ARs are not functionally different from their TTM homologs as long as their degree of activation leads to similar $[cAMP]_i$ responses. However, the above experiments do not exclude possible differences in the compartmentalization of intracellular cAMP cascade when cAMP is synthesized upon OSM or TTM $\beta$-AR stimulation. To obtain further insight, we compared the effect of PEG-Iso and Iso on PKA activity in the bulk cytoplasm and in the nucleus. For that, we used genetically encoded A-kinase activity FRET-based reporters (AKAR3)

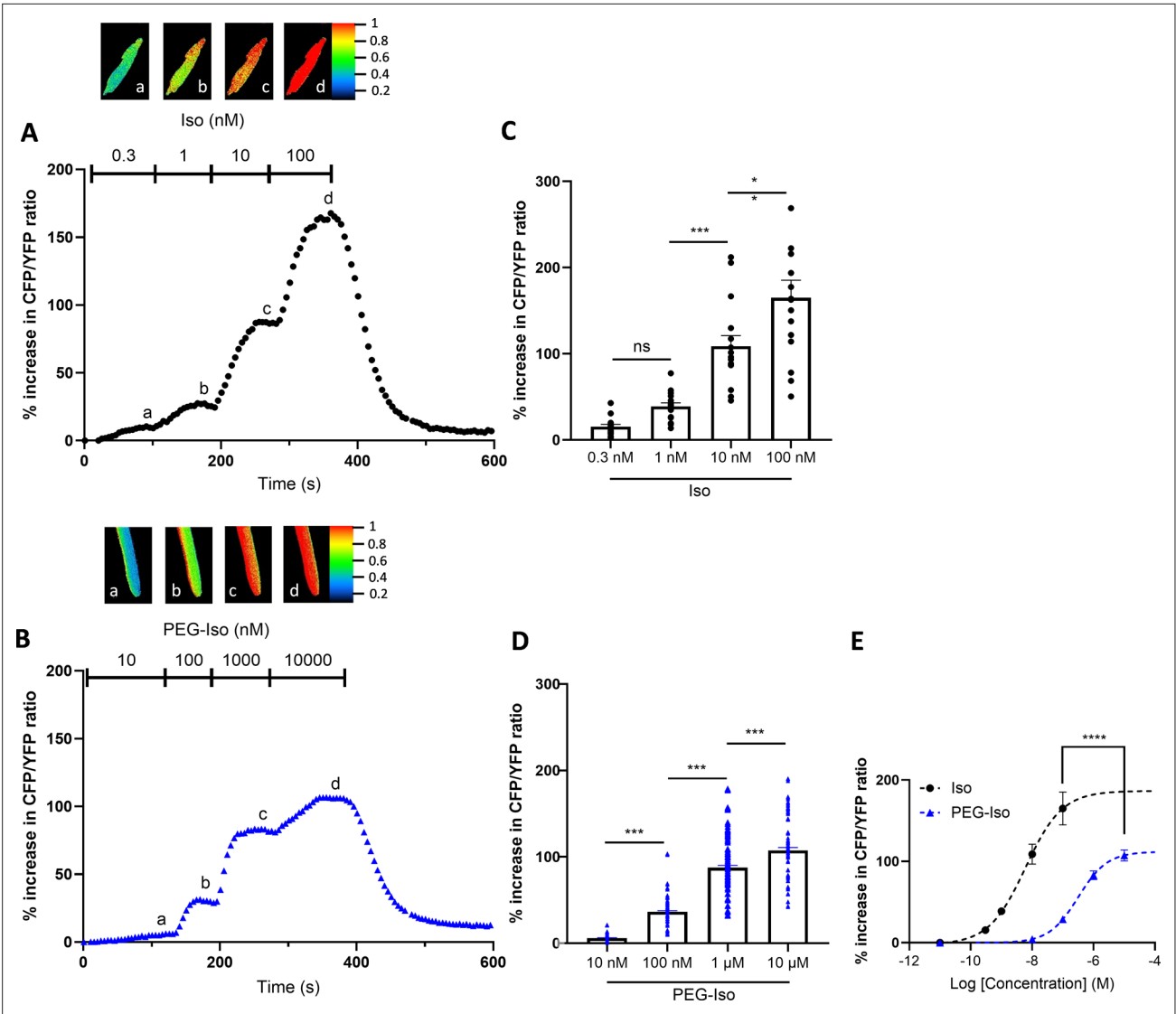

**Figure 3.** Effects of Iso and PEG-Iso on cytosolic cAMP in adult rat ventricular cardiomyocytes. Freshly isolated adult rat ventricular myocytes (ARVMs) were infected with an adenovirus encoding the Epac-S$^{H187}$ FRET-based cytosolic cAMP sensor for 48 hr at 37°C at a multiplicity of infection of 1000 pfu/cell. (**A**) Typical experiment showing the time course of CFP/YFP ratio during successive applications of four increasing Iso concentrations: 0.3, 1, 10, and 100 nM. (**B**) Similar experiment showing the time course of CFP/YFP ratio during successive applications of four increasing PEG-Iso concentrations: 10 and 100 nM, 1 and 10 µM. Pseudo-color images shown above the main graphs were taken at times indicated by the corresponding letters in the graphs. (**C, D**) Summary data from several similar experiments as in (**A**) and (**B**), respectively. The bars show the mean ± SEM of the data shown by symbols. Sixteen cells from 3 rats were used in (**C**); 32 cells from 4 rats in (**D**). One-way ANOVA and Tukey's multiple comparisons post hoc test were used: **p < 0.01; ***p < 0.001; ns, non-significant. (**E**) Comparison of the concentration–response curves in (**C**) and (**D**). A fit of the data to the Michaelis–Menten equation allowed us to estimate $E_{max}$ and $EC_{50}$ values for the effects of Iso and PEG-Iso, as well as 95% confidence intervals (CI 95%) for each parameter: $E_{max}$ = 168.9% (CI 95% 141.1, 196.8) and $EC_{50}$ = 4.6 nM (CI 95% 1.3, 7.9) for Iso; $E_{max}$ = 107.3% (CI 95% 98.1, 116.5) and $EC_{50}$ = 297.5 nM (CI 95% 174.1, 420.9) for PEG-Iso. The two curves are statistically different at p < 0.05 (*) since there was no overlap of CI 95% for each parameter.

targeted to these compartments by the addition of a nuclear export sequence (NES), and a nuclear localizing sequence (NLS), respectively (**Allen and Zhang, 2006**). As shown previously (**Bedioune et al., 2018**; **Haj Slimane et al., 2014**), adenoviral transfer allowed robust and compartment-specific expression of these biosensors after 24 hr in ARVMs. In the experiments shown in **Figure 7**, four increasing concentrations of Iso (0.3, 1, 3, and 10 nM: **Figure 7A, C**) and PEG-Iso (10, 30, 100 nM and 1 µM: **Figure 7B, D**) were successively applied to ARVMs infected with an AKAR3-NES probe, and cytosolic PKA activation was measured. Both compounds produced a clear concentration-dependent increase in cytosolic PKA activity, with a similar efficacy (maximal cytosolic PKA response of 59.7 ±

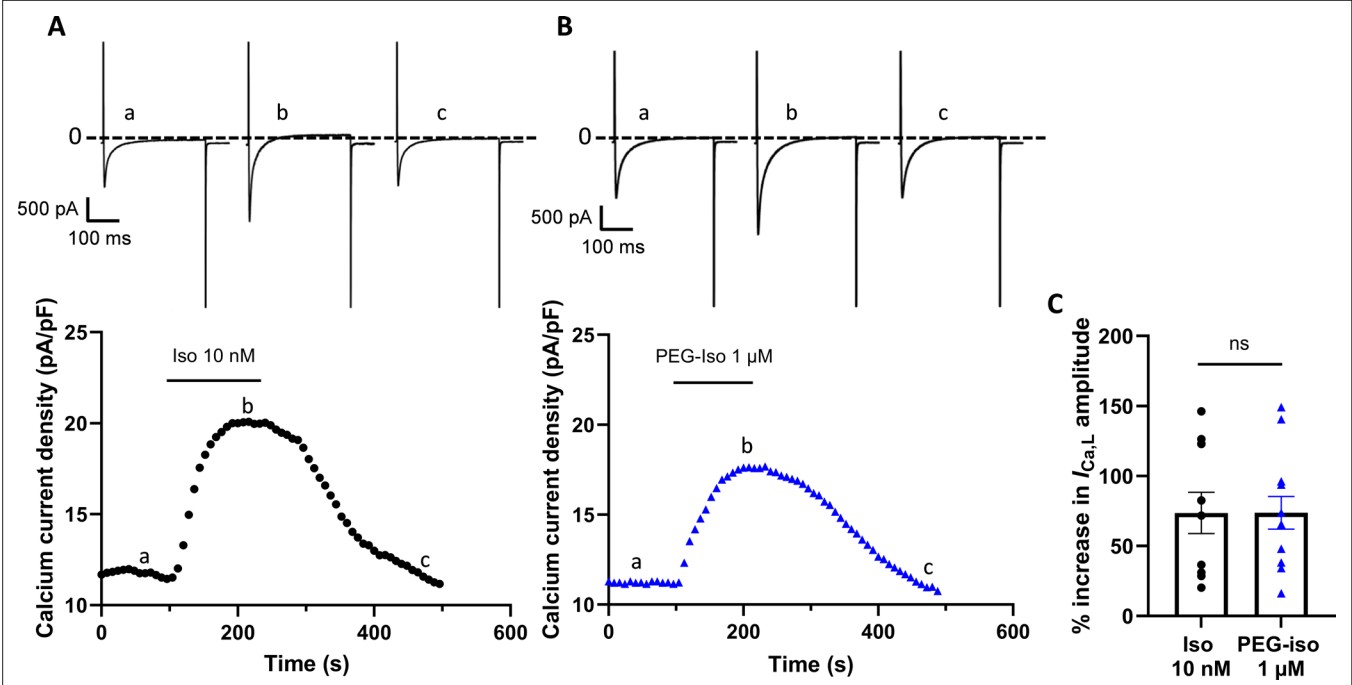

**Figure 4.** Comparison of the effects of PEG-Iso and Iso on $I_{Ca,L}$. The whole-cell patch-clamp technique was applied to adult rat ventricular myocytes (ARVMs) after 24 hr culture. (**A**) Typical experiment showing the time course of peak $I_{Ca,L}$ current density upon application of 10 nM Iso and during washout. (**B**) Similar experiment showing the response of peak $I_{Ca,L}$ current density to 1 μM PEG-Iso. Individual current traces shown above the graphs were taken at times indicated by the corresponding letters on the graphs. (**C**) Mean ± SEM. increase in peak $I_{Ca,L}$ in response to 10 nM Iso or 1 μM PEG-Iso stimulation. Student's unpaired $t$-test: ns, non-significant, $n$ = 11–12.

1.9% for Iso compared to 55.1 ± 3.9% for PEG-Iso; p = 0.33) but a ~100-fold lower potency for the effect of PEG-Iso as compared to Iso. However, when both nuclear cAMP (*Figure 8*) and nuclear PKA (*Figure 9*) activity was measured, clear differences between the effects of Iso and PEG-Iso were observed: While PEG-Iso and Iso still produced a concentration-dependent increase in nuclear cAMP/ PKA activity, the largest response to Iso (*Figures 8C and 9C*) was ≈2-fold greater compared to the largest response to PEG-Iso (*Figures 8D and 9D*; p = 0.002 (cAMP) and p = 0.033 (PKA), respectively). The same difference is observed when measuring the total nuclear protein phosphorylation by nuclear PKA (*Figure 9—figure supplement 1*) after stimulation with 10 nM Iso or 1 μM PEG-Iso by western blot.

*Figure 9—figure supplement 2* shows a plot of cytosolic and nuclear PKA activity as a function of cytosolic cAMP from the average data in the experiments in *Figures 3, 7, and 9* for three concentrations of PEG-Iso (10 and 100 nM and 1 μM) and Iso (0.3, 1, and 10 nM). It shows that for any given measured increase in [cAMP]$_i$, PEG-Iso is more efficient than Iso to increase PKA activity in the cytosol (*Figure 9—figure supplement 2A*), while on the contrary, Iso is more efficient than PEG-Iso to increase PKA activity in the nucleus (*Figure 9—figure supplement 2B*).

## Discussion

The main function of the TT system is to provide proximity between LTCCs in the TTM and RyR2 in the SR membrane (*Sipido and Cheng, 2013*; *Carl et al., 1995*). However, TTM also contains many other membrane proteins, such as receptors, enzymes, ion channels, and transporters. These include plasma membrane Ca²⁺-ATPase (*Brette and Orchard, 2003*), Na⁺–Ca²⁺ exchanger (*Scriven et al., 2000*; *Yang et al., 2002*), Na⁺ channels (*Scriven et al., 2000*), Na⁺/K⁺-ATPase (*Berry et al., 2007*), ACs (*Timofeyev et al., 2013*; *Gao et al., 1997*), and β-ARs (*Nikolaev et al., 2010*; *Gorelik et al., 2013*). The density of these membrane proteins is usually found to be higher in the TTM than in the external membrane. Hence, the TT network plays a determinant role in rapid activation and synchronous Ca²⁺

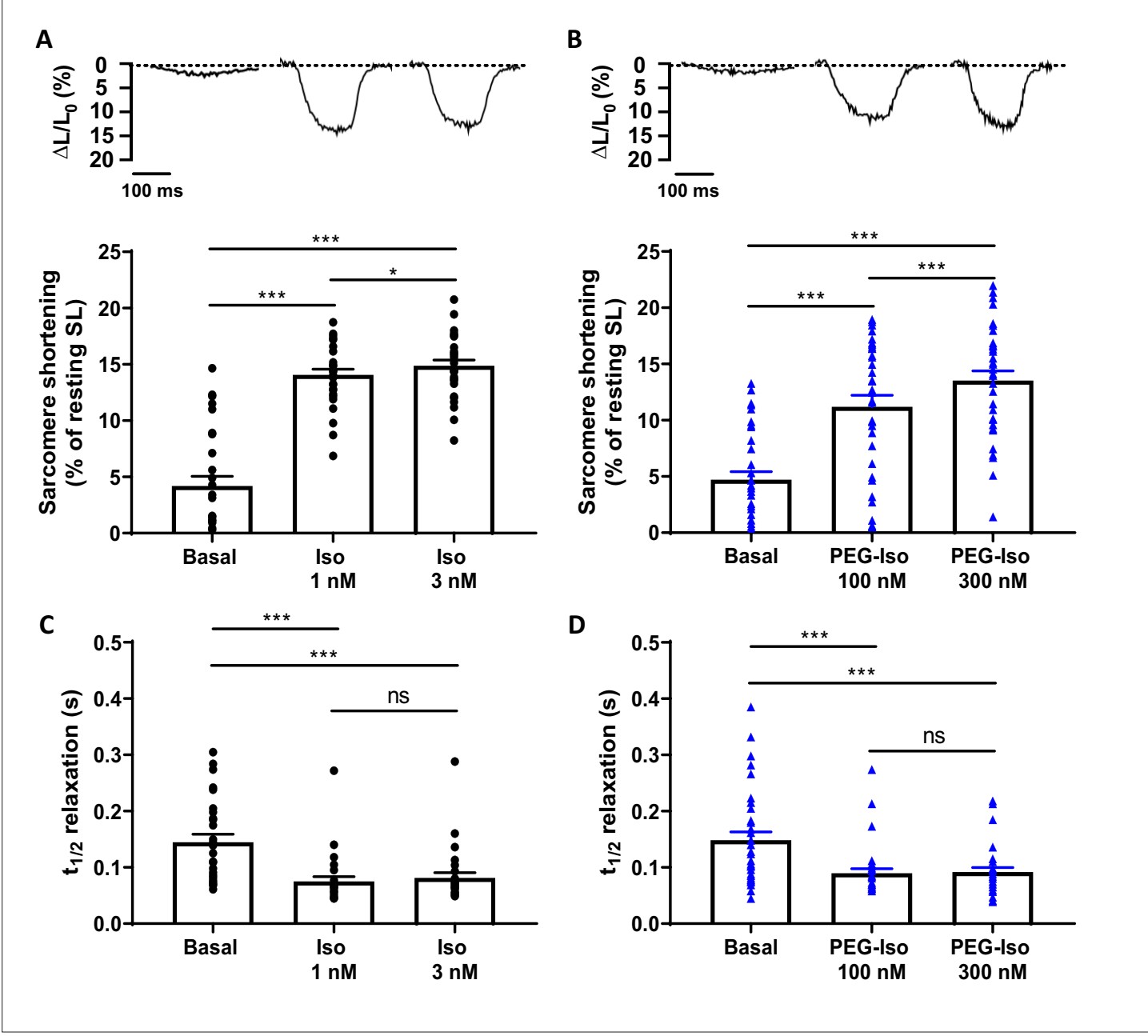

**Figure 5.** Comparison of the effects of PEG-Iso and Iso on sarcomere shortening. Representative traces of sarcomere shortening recorded in adult rat ventricular myocytes (ARVMs) paced at 0.5 Hz and loaded with Fura-2 AM 1 μM showing the effects of Iso 1 and 3 nM (**A**) and PEG-Iso 100 and 300 nM(**B**). The bars in (**A**) and (**B**) show the mean ± SEM of the data shown by symbols. (**C, D**) Average time-to-50% relaxation of sarcomere shortening from experiments shown in (**A**) and (**B**), respectively. Thirty cells from 4 rats were used in (**A**) and (**C**); 34 cells from 4 rats in (**B**) and (**D**). One-way ANOVA and Tukey's post hoc test: *p < 0.05; ***p < 0.001; ns, non-significant.

release, cellular signaling and, consequently, cardiac contraction in both human and animal models (*Song et al., 2006*; *Soeller and Cannell, 1999*; *Hatano et al., 2012*).

Any given membrane protein is likely to serve different functions and be regulated in a different manner whether it is located in TTM or in OSM. Attempts to address this important question have so far been based on the elimination of the TT network (detubulation) using a hyperosmotic shock with molar concentrations of formamide (*Brette et al., 2006*; *Moench et al., 2013*; *Cros and Brette, 2013*; *Brette et al., 2005*; *Fowler et al., 2004*; *Brette et al., 2004*; *Thomas et al., 2003*; *Despa et al., 2003*; *Brette et al., 2002*; *Kawai et al., 1999*). This technique, first used in skeletal muscle

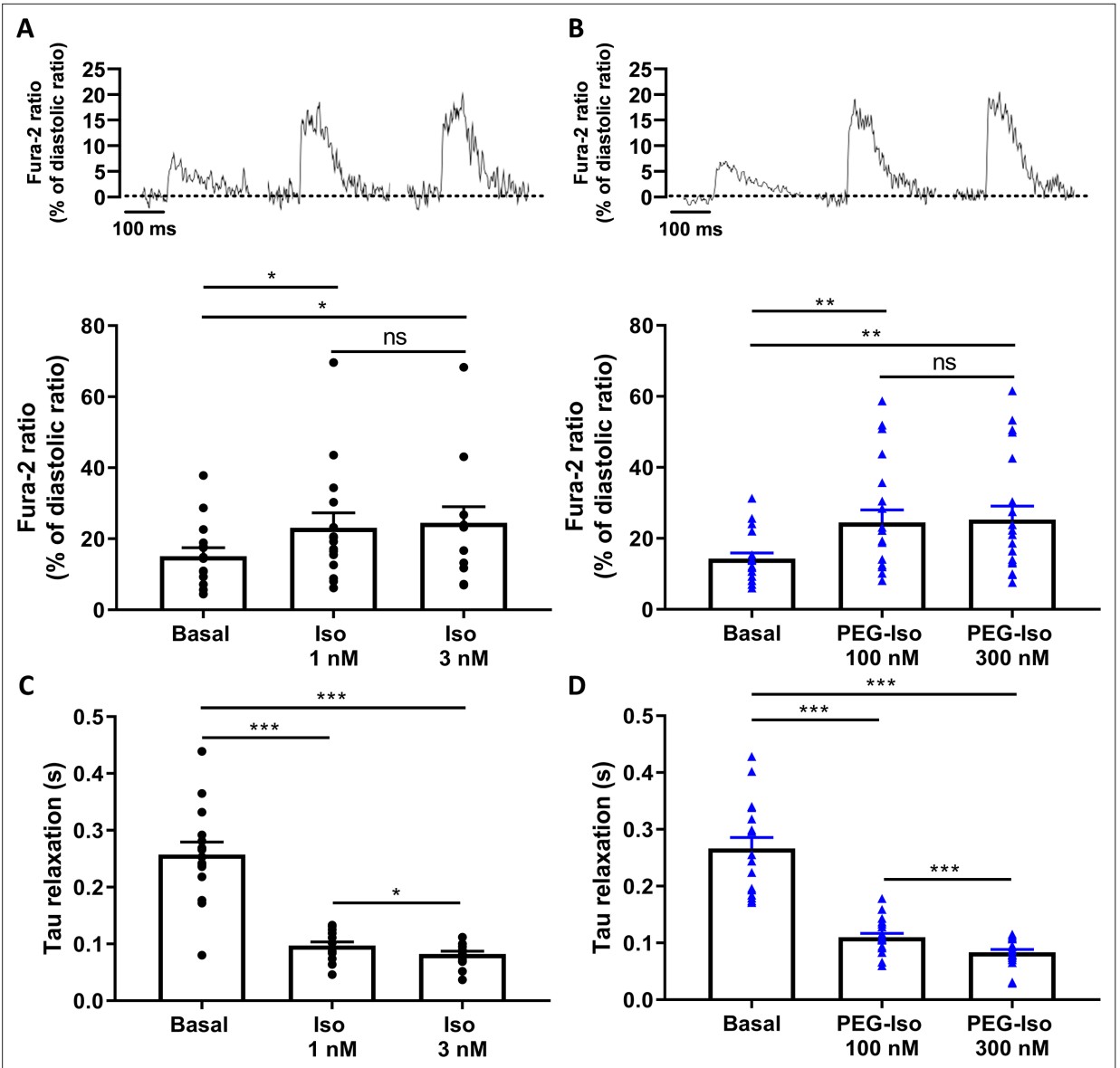

**Figure 6.** Comparison of the effects of PEG-Iso and Iso on $Ca^{2+}$ transients. Representative traces of $Ca^{2+}$ transients recorded in adult rat ventricular myocytes (ARVMs) paced at 0.5 Hz and loaded with Fura-2 AM (1 µM) showing the effect of Iso 1 and 3 nM (**A**) and PEG-Iso 100 and 300 nM (**B**). The bars in (**A**) and (**B**) show the mean ± SEM with individual data points denoted by symbols. (**C, D**) Exponential time constant (*Tau*) of relaxation of $Ca^{2+}$ transients from experiments shown in (**A**) and (**B**), respectively. Thirty cells from 4 rats were used in (**A**) and (**C**); 34 cells from 4 rats in (**B**) and (**D**). One-way ANOVA and Tukey's post hoc test: *p < 0.05; **p < 0.01; ***p < 0.001; ns, non-significant.

(*Argiro, 1981*), was introduced in the cardiac field by *Kawai et al., 1999* and provided an excellent experimental tool for numerous T-tubular studies. However, the potential role of shock-induced detubulation in physiologically and pathophysiologically relevant conditions is essentially unknown. The method is very harsh on the cells with fewer than 10% of the cells surviving the procedure, which raises issues about whether the 'survivors' are truly representative cells (*Bourcier et al., 2019*). Furthermore, osmotic shock produces a sealing of the TTs which remain present as vesicles inside the cell (*Moench et al., 2013*) and may still respond to hormonal or pharmacological challenges and signaling cascades even if they are electrically disconnected from the surface membrane (*Moench and Lopatin, 2014*; *Scardigli et al., 2024*).

To overcome these limitations, we introduce here a completely different approach based on size exclusion. We proposed to enlarge the size of a ligand molecule by attaching it to a 5-kDa chain of PEG so that it does not access TTM but remains active on OSM. PEG is a hydrophilic, flexible, and

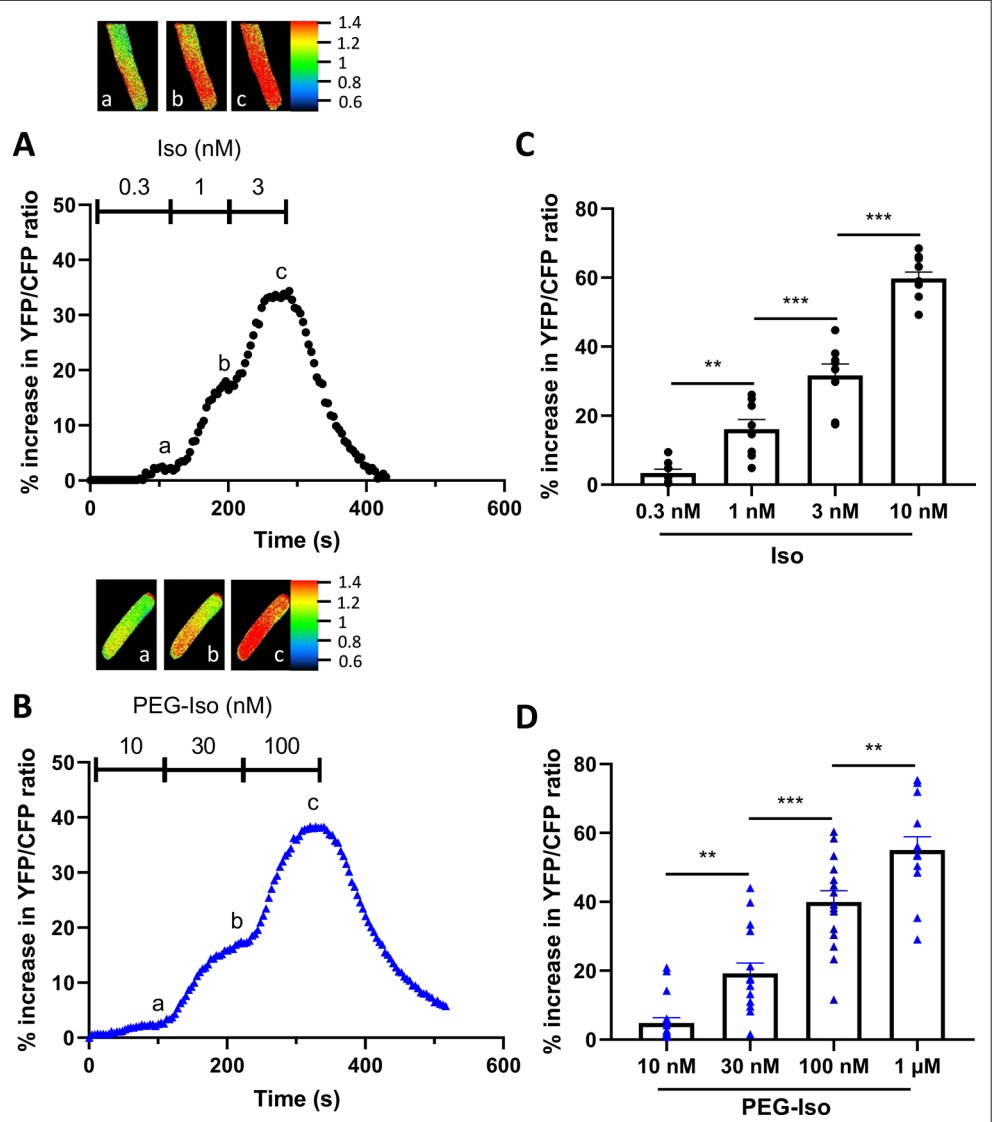

**Figure 7.** Comparison of the effects of PEG-Iso and Iso on cytosolic PKA activity. Freshly isolated adult rat ventricular myocytes (ARVMs) were infected with an adenovirus encoding the AKAR3-NES FRET-based PKA sensor for 48 hr at 37°C at a multiplicity of infection of 1000 pfu/cell. (**A**) Typical experiment showing the time course of YFP/CFP ratio during successive applications of three increasing Iso concentrations: 0.3, 1, and 3 nM. (**B**) Similar experiment showing the time course of YFP/CFP ratio during successive applications of three increasing PEG-Iso concentrations: 10, 30, and 100 nM. Pseudo-color images shown above the main graphs were taken at times indicated by the corresponding letters in the graphs. (**C, D**) Summary data from several similar experiments as in (**A**) and (**B**), respectively. The bars show the mean ± SEM of the data shown by symbols. Three rats and 8–10 cells were used in (**C**); 3–5 rats and 13–18 cells in (**D**). One-way ANOVA and Tukey's multiple comparisons post hoc test: **p < 0.01; ***p < 0.001.

rather inert polymer. The covalent linking of one or several PEG chains to a therapeutic molecule, called PEGylation (**Veronese and Harris, 2002**), is commonly used in several products on the market (**Pasut and Veronese, 2012**). PEGylated molecules exhibit improved biodistribution and pharmacokinetics, better stability and solubility, reduced immunogenicity, and longer plasma half-life due to both reduced renal filtration and proteolysis, when compared to non-PEGylated analogs (**Abuchowski et al., 1977**; **Harris and Chess, 2003**). In our study, we diverted the same technology toward an entirely different objective: Instead of using PEGylated molecules to improve drug formulation, pharmacokinetics, and efficacy, we propose to use PEGylated drugs as *key holders* to prevent drug (*key*) access in the TT network and thus limit its access to the outer surface of the cell. We provide a proof

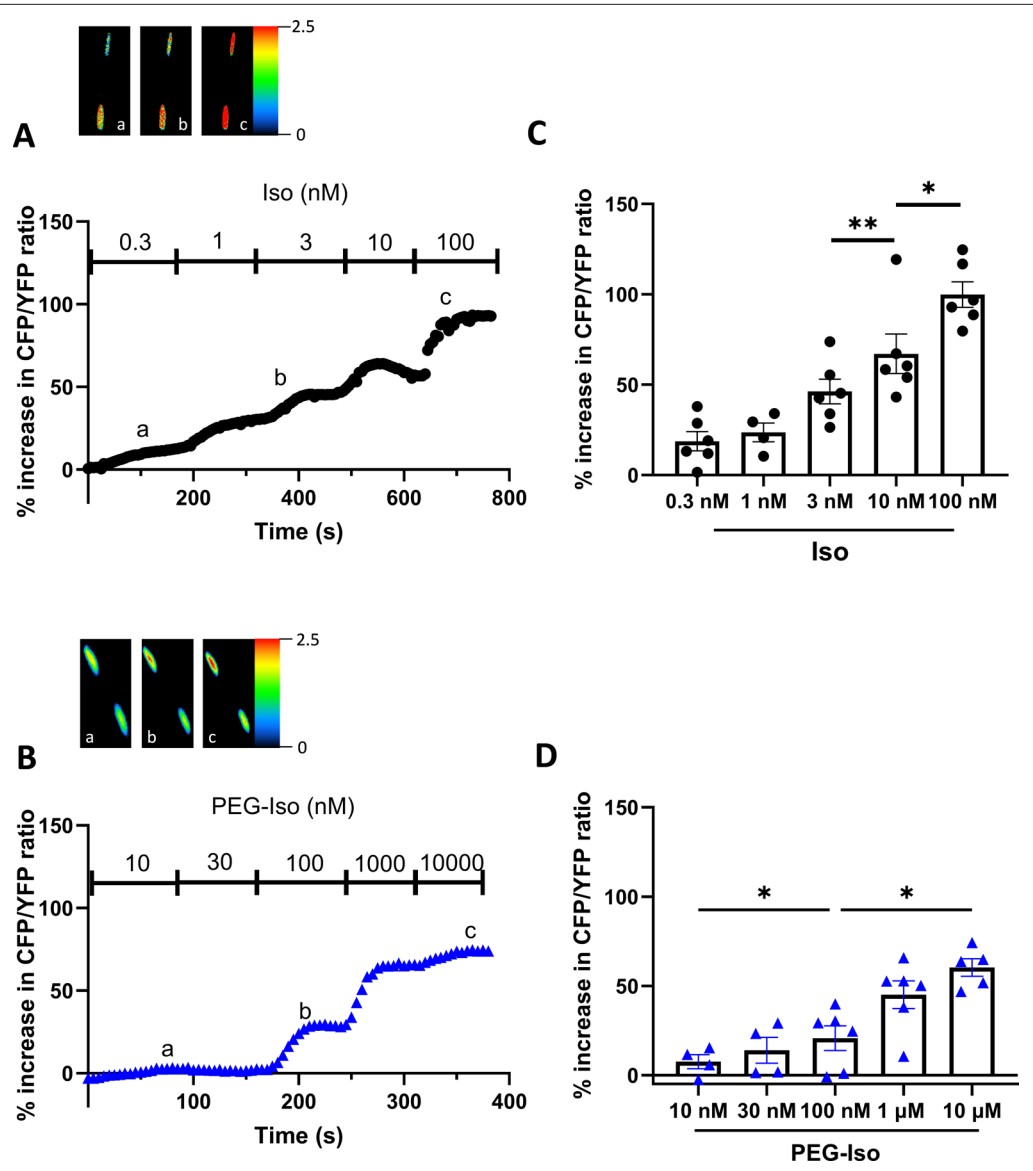

**Figure 8.** Comparison of the effects of PEG-Iso and Iso on nuclear cAMP activity. Freshly isolated adult rat ventricular myocytes (ARVMs) were infected with an adenovirus encoding the nuclear Epac-SH187-3NLS FRET-based nuclear cAMP sensor for 48 hr at 37°C at a multiplicity of infection of 1000 pfu/cell. (**A**) Typical experiment showing the time course of CFP/YFP ratio during successive applications of increasing Iso concentrations and (**B**) a dose–response of increasing PEG-Iso concentrations. Pseudo-color images shown above the main graphs were taken at times indicated by the corresponding letters in the graphs. (**C, D**) Summary data from several similar experiments as in (**A**) and (**B**), respectively. The bars show the mean ± SEM of the data shown by symbols. Two rats and 6 cells were used in (**C**) and 6 cells from 3 rats in (**D**). One-way ANOVA and Tukey's multiple comparisons post hoc test were used: *$p < 0.05$; **$p < 0.01$.

of concept that this approach works using PEG$_{5000}$ functionalized with non-permeant FITC (PEG-FITC): When ARVMs were exposed to PEG-FITC, fluorescence was only seen on the periphery of the cell, while when the non-PEGylated free diffusible dye was used, fluorescence was seen in the TT compartment (*Figure 1*). Thus, large molecular weight PEGs are prevented from diffusing within TTs, whereas small molecular weight fluorophores can easily enter.

This finding was surprising, though, since longitudinal TT diameter from healthy control ARVMs varies from 50 to 350 nm (~200 nm on average; *Wagner et al., 2012*; *Soeller and Cannell, 1999*; *Wagner et al., 2012*), which is significantly larger than the size of a PEG$_{5000}$ molecule, which is ≈5 nm.

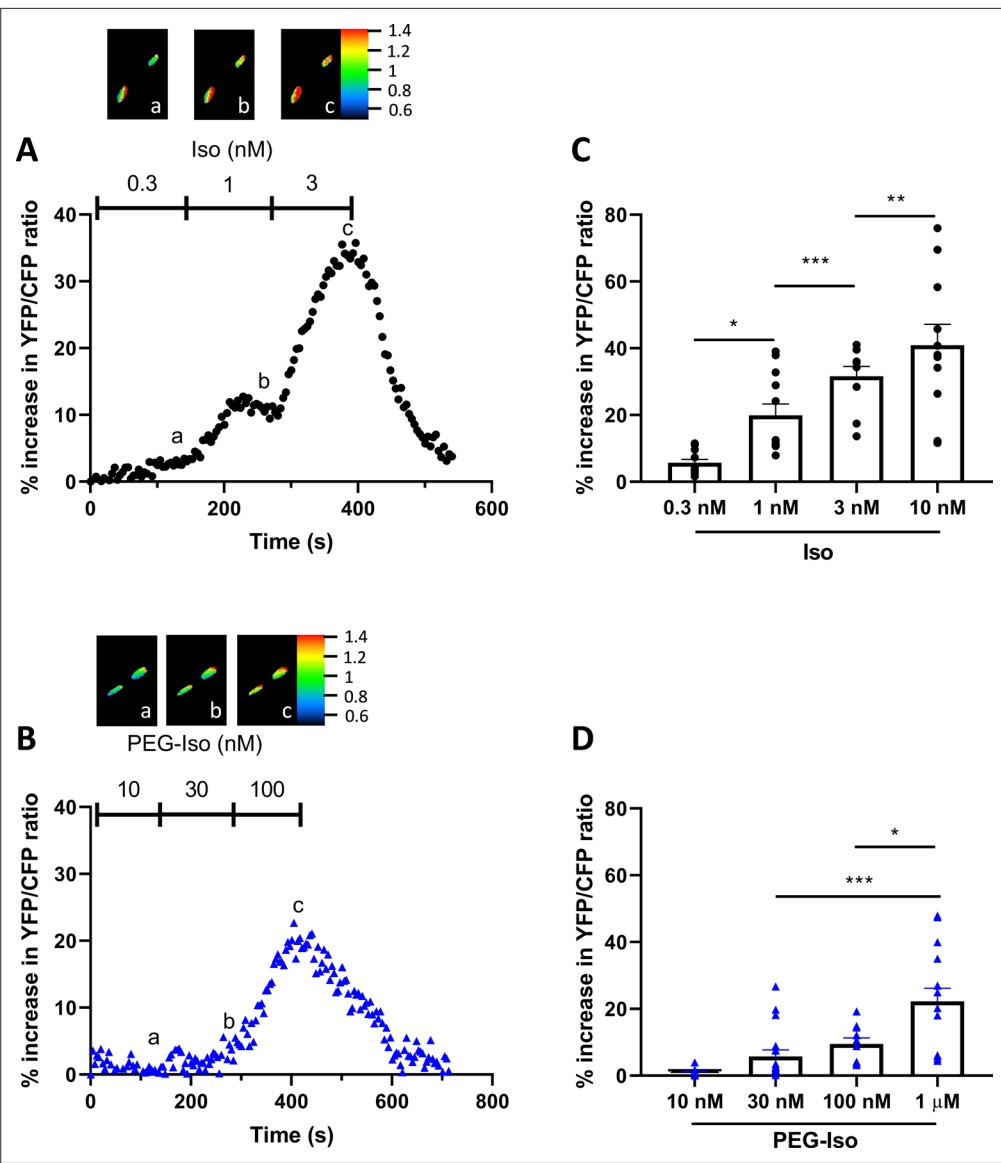

**Figure 9.** Comparison of the effects of PEG-Iso and Iso on nuclear PKA activity. Freshly isolated adult rat ventricular myocytes (ARVMs) were infected with an adenovirus encoding the AKAR3-NLS FRET-based PKA sensor for 48 hr at 37°C at a multiplicity of infection of 1000 pfu/cell. (**A**) Typical experiment showing the time course of YFP/CFP ratio during successive applications of three increasing Iso concentrations: 0.3, 1, and 3 nM. (**B**) Similar experiment showing the time course of YFP/CFP ratio during successive applications of three increasing PEG-Iso concentrations: 10, 30, and 100 nM. Pseudo-color images shown above the main graphs were taken at times indicated by the corresponding letters in the graphs. (**C, D**) Summary data from several similar experiments as in (**A**) and (**B**), respectively. The bars show the mean ± SEM of the data shown by symbols. Three rats and 10–12 cells were used in (**C**) 3–5 rats and 10–17 cells in (**D**). One-way ANOVA and Tukey's multiple comparisons post hoc test: *$p < 0.05$; **$p < 0.01$; ***$p < 0.001$.

The online version of this article includes the following source data and figure supplement(s) for figure 9:

**Figure supplement 1.** Comparison of the effects of PEG-Iso and Iso on nuclear proteins phosphorylation levels by PKA.

**Figure supplement 1—source data 1.** TIF files containing original western blots for *Figure 9—figure supplement 1A, B*.

**Figure supplement 1—source data 2.** PRISM, PDF, and Excel files for the western blot analysis in *Figure 9—figure supplement 1A–C*.

*Figure 9 continued on next page*

*Figure 9 continued*

**Figure supplement 2.** Plot of cytosolic and nuclear PKA activity as a function of cytosolic cAMP in response to Iso and PEG-Iso.

**Figure supplement 3.** PEG-Iso cytosolic cAMP and nuclear cAMP/PKA responses after treatment of cells with neuraminidase.

**Figure supplement 4.** Detubulation normalizes the effect of Iso and PEG-Iso.

**Figure supplement 5.** Effect of permeant (propranolol) and impermeant (sotalol) β-blockers on nuclear PKA response to β-AR stimulation by Iso.

**Figure supplement 6.** T-tubule network is still robust up to 48 hr post-isolation.

However, a number of recent studies have shown that solute movement in the TT network is strongly restricted even though the diameter of TTs is far larger than the molecular dimensions of typical solutes (*Entcheva, 2018*; *Hong et al., 2014*; *Kong et al., 2018*; *Scardigli et al., 2017*; *Uchida and Lopatin, 2018*). Impediment of solute movement in the TT network was proposed to be caused by the presence of the glycocalyx (*Langer et al., 1982*) at the cell surface, reducing the effective diameter of a TT (*Parfenov et al., 2006*; *Hong et al., 2014*; *Kong et al., 2018*). This was supported by two findings. First, that pre-treatment of isolated ARVMs with neuraminidase, an enzyme which cleaves sialic acid from oligosaccharide chains in the glycoprotein matrix on the external cell surface, allowed PEG-FITC to enter the TT compartment (*Figure 1*). Second, when ARVMs pre-treated with neuraminidase were stimulated with PEG-Iso, the FRET-derived responses were much larger than in the non-treated cells, indicative of increased access to the β-ARs in the TTM, similar to free Iso (*Figure 9—figure supplement 3*). The cytosolic cAMP response increased by ≈100% (*Figure 9—figure supplement 3A*), while nuclear cAMP (*Figure 9—figure supplement 3C*) and nuclear PKA responses (*Figure 9—figure supplement 3E, G*) were ≈50% greater. Controls using Iso confirmed this to be an effect of the glycocalyx being degraded rather than a direct effect of neuraminidase on the β-ARs (*Figure 9—figure supplement 3B, D, F*). Thus, when the protocol used for cell isolation is sufficiently gentle to preserve the integrity of the extracellular matrix in the TT, which must be the case in our experimental conditions, it is possible to prevent a ligand from accessing the TTM if the size of the ligand molecule is sufficiently enlarged. PEG molecules of 5000 Da MW appear to provide a good compromise between solubility issues and size exclusion capability.

Several studies have shown localization of β-ARs particularly β₂-ARs (*Wright et al., 2014*; *Wright et al., 2018*) in caveolae. Our hypothesis is that, unlike what is occurring within TT, caveolae are not filled with a dense extracellular matrix, possibly because of a better access to enzymatic digestion during cell isolation. Future experiments will test this hypothesis by disrupting caveolae with methyl-beta-cyclodextrin (*Calaghan and White, 2006*) to see how this affects the response to PEG-Iso.

PEG-Iso was thus synthesized by linking Iso to PEG₅₀₀₀ molecules. Characterization of its binding properties showed that PEG-Iso binds to β₁- and β₂-ARs but with ~2 orders of magnitude lower affinity than Iso (*Figure 2*). Since PEG was grafted on the lateral amino group of Iso (*Figure 1—figure supplement 1*), a position shown to maintain its affinity for β-ARs (*Ruffolo et al., 1995*) and away from the active pharmacophore for β-AR binding, it was unlikely that the chemical modification per se was the reason for the decreased affinity. We confirmed that this was the case by detubulating cells with 300 μM imipramine (*Bourcier et al., 2019*). Without the TT network, only the OSM β-ARs were available for stimulation. Under these conditions, affinity-adjusted Iso and PEG-Iso had identical effects on cytosolic and nuclear cAMP/PKA stimulation (*Figure 9—figure supplement 4*). We also addressed the possibility that PEG could change the membrane permeability of Iso and differentially stimulate intracellular β-ARs, the existence of which has been proposed by other groups (*Wang et al., 2021*). We found that irrespective of whether we used a membrane-permeable (propranolol) or membrane-impermeable (sotalol) β-AR antagonist, the cAMP stimulatory effects of Iso could be fully inhibited (*Figure 9—figure supplement 5*), and so under our conditions, the stimulatory effects of Iso are confined to the cell membrane. Therefore, the only difference between Iso and PEG-Iso is the ability to bind the β-ARs and not an effect of PEG itself.

We thus explored a possible involvement of the PEG conformation in this phenomenon using molecular dynamics simulations. Because of its hydrophilicity and flexibility, PEG is very dynamic in solution and constantly changes conformation. Consequently, the Iso moiety in PEG-Iso is rarely fully

accessible to solvent and spends most of the time with the PEG chain wrapped around it (*Figure 2*). Based on the calculations, only ~1% of PEG-Iso conformations are able to bind β-ARs as compared to 100% for free Iso, which is in the order of magnitude of the measured difference in their respective binding affinity on β-ARs.

When applied on ARVMs, PEG-Iso produced stimulatory effects on $[cAMP]_i$, PKA activity, $I_{Ca,L}$, sarcomere shortening, and $Ca^{2+}$ transients, which all had the hallmarks of a β-AR stimulation (*Figures 3–9*). Because of the reduced binding affinity toward β-ARs, PEG-Iso produced its effect at ~2 orders of magnitude larger concentrations than Iso. But there were two other striking differences between the effects of PEG-Iso and Iso which strongly suggest that the two ligands did not act on the exact same populations of receptors.

First, PEG-Iso produced a stimulation of $[cAMP]_i$ with a much lower efficacy than Iso (*Figure 3*). The simplest explanation for this result is that PEG-Iso activates only β-ARs located in the OSM, while Iso activates β-ARs located in both OSM and TTM. Since both populations of β-ARs are likely coupled to ACs (*Timofeyev et al., 2013*; *Gao et al., 1997*), one would expect an activation of OSM β-ARs with PEG-Iso to lead to a smaller maximal increase in $[cAMP]_i$ than an activation of all β-ARs with Iso. This diminution is in line with decreased cAMP production sites by detubulation (*Bourcier et al., 2019*). Cyclic AMP must emanate mainly from sarcolemmal $β_1$-ARs, which produce highly diffusible signals throughout the cell in contrast to $β_2$-ARs (*Nikolaev et al., 2006*). However, there are conflicting results regarding the distribution of β-ARs on the cardiac cell membrane. Immunohistochemical data have suggested that $β_1$- and $β_2$-ARs are present in OSM and TTM in mouse heart (*Zhou et al., 2000*). Using radioligand binding, $β_1$-AR density was found to be almost twofold more concentrated in OSM than TTM, whereas $β_2$-AR density was evenly distributed across the entire cell surface in dog heart (*He et al., 2005*). Using nanoscale live-cell scanning ion conductance and fluorescence resonance energy transfer microscopy techniques in healthy ARVMs, Nikolaev et al. found that $β_1$-ARs are distributed across the entire cell surface while $β_2$-ARs are localized exclusively in TTM (*Nikolaev et al., 2010*). While the distribution of $β_1$-ARs across the entire cell surface was confirmed in ARVMs in another study using formamide detubulation, $β_2$-ARs were found to be only present in OSM (*Cros and Brette, 2013*). Additional experiments using PEGylated selective $β_1$- and $β_2$-AR agonists and antagonists are needed to solve this issue and to precisely characterize the role of each β-AR subtype in the different membrane compartments.

Second, PEG-Iso produced a much lower stimulation of nuclear PKA activity than Iso (*Figure 9*) even though both ligands were equally efficient in stimulating cytosolic PKA activity (*Figure 7*). Also, when cytosolic and nuclear PKA activity measured during PEG-Iso or Iso stimulation were plotted as a function of $[cAMP]_i$ measured under the same conditions, PEG-Iso was found to be more efficient than Iso to increase PKA activity in the cytosol, while on the contrary, Iso is more efficient than PEG-Iso to increase PKA activity in the nucleus (*Figure 9—figure supplement 2*). A likely interpretation of these results is that β-ARs located in TTM (or a β-AR subtype more concentrated in TTM) are more efficiently coupled to nuclear PKA than those located in OSM, while β-ARs located in OSM (or a β-AR subtype more concentrated in OSM) are more efficiently coupled to cytosolic PKA. In a recent study, we showed that $β_2$-ARs, unlike $β_1$-ARs, are inefficient in activating nuclear PKA activity in ARVMs (*Bedioune et al., 2018*). Thus, if the ratio of $β_2/β_1$ receptors is larger in OSM than in TTM, as discussed above (*Cros and Brette, 2013*), then this would explain the lower efficacy of PEG-Iso to activate nuclear PKA activity as compared to Iso. Recent studies have shown that perinuclear PKA has the ability to activate without detachment of the catalytic subunits, and thus to phosphorylate its targets in close proximity (*Smith et al., 2013*; *Smith et al., 2017*). According to these studies, only supra-physiological concentrations of cAMP could lead to dissociation of the catalytic subunits and their translocation into the nucleus. Perinuclear PKA is anchored to the nuclear membrane via its interaction with mAKAP and its dynamics are finely regulated by AC5 and PDE4D3 (*Diviani et al., 2011*), AC5 being concentrated in TTM (*Timofeyev et al., 2013*) and PDE4D3 being linked to mAKAP (*Dodge-Kafka et al., 2005*). The location of AC5 may therefore be crucial for nuclear signaling and explain why β-ARs present in TTM are necessary for activation of nuclear PKA. One major limitation of our data is that there is a reduction in TTs over the ≈36-hr infection period needed for robust FRET sensor expression (*Figure 9—figure supplement 6*). Therefore, the differences between OSM and TTM may be even greater than we have observed with respect to nuclear PKA activation.

Another important finding from this study concerns the contribution of OSM β-ARs to the regulation of EC coupling. Activation of β-ARs only in OSM with PEG-Iso produced a similar increase in $I_{Ca,L}$ amplitude, sarcomere shortening, and $Ca^{2+}$ transients as Iso, when both ligands were used at concentrations producing an equivalent elevation of $[cAMP]_i$. Since OSM contributes to ~60% of total cell membrane in ARVMs (*Bourcier et al., 2019*), either β-ARs and ACs are more concentrated in OSM than TTM, or they are in large excess over what is needed to activate PKA phosphorylation of proteins involved in EC coupling. It would also suggest that small GTPase Rab proteins mediated increases in $Ca_V1.2$ trafficking to the TTM that occur in response to β-AR activation (*Del Villar, 2021*) are sufficiently activated by OSM cAMP alone for a similar increase in $I_{Ca,L}$ amplitude to have been observed. Also, cAMP produced at OSM must diffuse rapidly in the cytosol in order to activate PKA phosphorylation of substrates located deep inside the cell, such as LTCCs in TTM. A recent study showed that under basal conditions, cAMP is bound to proteins and its diffusion in the cytosol is slow; but when cAMP increases in the cytosol, for example as a result of β-AR stimulation, the cAMP-binding sites become saturated and cAMP diffuses freely (*Bock et al., 2020*).

While this study provides new insight on the differential function of OSM vs. TTM β-ARs in healthy cardiomyocytes raise obvious questions on how their role is altered in cardiac pathology. A reduction in the density and a disorganization of the TT network is frequently observed in pathological conditions (*Guo et al., 2013*; *Wagner et al., 2012*; *Lawless et al., 2019*), with a substantial impact on both EC coupling and contractile function (*Sipido and Cheng, 2013*). A previous study showed a redistribution of $β_2$-ARs from TTM to OSM in failing ARVMs (*Nikolaev et al., 2010*). As β-ARs are not simply bystanders but also participate in the remodeling during pathological hypertrophy and failure, it is important to know how the OSM vs. TTM subpopulations of β-AR subtypes contribute to this process. The PEGylation technology should enable elucidation of the changes that occur in OSM vs. TTM distribution of β-ARs in ventricular cells isolated from failing hearts, and to determine how this impacts on cellular function. In conclusion, this study provides a novel approach to distinguish the function of a membrane protein depending on its location on the cell membrane. Whereas we focused here on the function of β-ARs, the size exclusion strategy provided by ligand PEGylation can be extended to other ligands, such as selective agonists or antagonists of β-AR subtypes or other G-protein-coupled receptors, dihydropyridine agonists or antagonists of LTCCs. More generally, it should pave the road to exploring and comparing the function of any membrane receptor, channel, transporter, or enzyme in TTM and OSM compartments in intact cardiomyocytes.

## Materials and methods

### Animals

All animal care and experimental procedures were performed in accordance with the ARRIVE guidelines and conform to the European Community guiding principles in the Care and Use of Animals (Directive 2010/63/EU of the European Parliament), the local Ethics Committee (CREEA Ile-de France Sud) guidelines, and the French decree no. 2013-118 of February 1, 2013 on the protection of animals used for scientific purposes (JORF no. 0032, February 7, 2013, p2199, text no. 24). Animal experiments were carried out according to the European Community guiding principles in the care and use of animals (2010/63/UE), the local Ethics Committee (CREEA Ile-de-France Sud) guidelines and the French decree no. 2013-118 on the protection of animals used for scientific purposes. Adult male rats (*Rattus norvegicus*) belonging to the Wistar strain (Janvier, Le Genest St Isle, France) have been used. This is a versatile strain of albino rats, selected by Donaldson in 1906 at the Wistar Institute (USA).

### Synthesis of PEG-Iso

Isoprenaline was grafted into the reactive end of 5000 Da PEG ($PEG_{5000}$). A carbodiimide reaction was carried out between the carboxylic acid of PEG and the amine function of Iso (*Figure 1—figure supplement 1*). In a 25-ml amber vial, 41.3 mg of DCC (0.2 mmol; 2eq), 23 mg of NHS (0.2 mmol; 2eq), and 500 mg of OH-$PEG_{5000}$-COOH (0.1 mmol; 1eq) were solubilized in 10 ml of dichloromethane. 1 ml of triethylamine was added to the syringe. The solution was stirred for 5 hr at RT. 21.1 mg isoprenaline (0.1 mmol; 1 eq) was added to the reaction mixture. The mixture was left to stir overnight. The solution was then filtered through silica gel and precipitated twice in cold diethyl ether. After evaporation of the ether, the final powder was solubilized in $H_2O$.

## Purification of PEG-Iso

A purification step was introduced to obtain a pure PEG-Iso product and to eliminate the various by-products obtained, potential impurities, and especially traces of ungrafted Iso which could interfere in the biological experiments. A HPLC 1290 Infinity II (Agilent Technologies) consisting of a 4-channel binary pump, UV/visible detector (PDA diode array), 1260 Infinity DEDL light scattering detector, and C18 XBridge column, 4.6 × 150 mm, 5 µm (17 ml/min flow rate) was used. The mobile phase consisted of a mixture of water +0.1% formic acid and acetonitrile. This phase was pushed according to a gradient from 1 to 100% in 15 min into the stationary phase for the purification of PEG-Iso. Detection was carried out in the UV at 254 and 280 nm. PEG-Iso was solubilized in water. Several successive injections of 300 µl each were performed. The retention time of PEG-Iso was 11 min. The fractions containing PEG-Iso were recovered, evaporated, and then frozen and freeze-dried. Finally, the powder obtained was stored at –20°C and protected from light.

## Cardiomyocyte isolation and culture

Adult male Wistar rats (250–300 g) were anesthetized by intraperitoneal injection of pentobarbital (0.1 mg/g) and hearts were excised rapidly. Individual adult rat ventricular cardiomyocytes were obtained by retrograde perfusion of the heart as previously described (*Mika et al., 2013*). For enzymatic dissociation, the hearts were perfused during 5 min at a constant flow of 6 ml/min at 37°C with a $Ca^{2+}$-free Ringer solution containing (in mM): NaCl 117, KCl 5.7, $NaHCO_3$ 4.4, $KH_2PO_4$ 1.5, $MgCl_2$ 1.7, D-glucose 11.7, $Na_2$-phosphocreatine 10, taurine 20, and 4-(2-hydroxyethyl)piperazine-1-ethanesulfonic acid (HEPES) 21, pH 7.1. This was followed by a perfusion at 4 ml/min for 40 min with the same solution containing 1 mg/ml of collagenase A (Roche Diagnostics GmbH, Mannheim, Germany) plus 300 µM ethylene glycol tetraacetic acid (EGTA) and $CaCl_2$ to adjust free $Ca^{2+}$ concentration to 20 µM. The ventricles were then separated, finely chopped, and gently agitated to dissociate individual cells. The resulting cell suspension was filtered on gauze, and the cells were allowed to settle down. The supernatant was discarded and cells were resuspended four more times with Ringer solution at increasing $[Ca^{2+}]$ from 20 to 300 µM. Freshly isolated cells were suspended in minimal essential medium (MEM: M 4780; Sigma, St. Louis, MO, USA) containing 1.2 mM $[Ca^{2+}]$ supplemented with 2.5% fetal bovine serum (FBS, Invitrogen, Cergy-Pontoise, France), 1% penicillin–streptomycin, 20 mM HEPES (pH 7.6), and plated on 35 mm, laminin-coated (10 mg/ml) culture dishes at a density of $10^4$ cells per dish and kept at 37°C (5% $CO_2$). After 1 hr, the medium was replaced by 300 µl of FBS-free MEM. To perform FRET imaging, the medium was replaced by 300 µl of FCS-free MEM or transduced with an adenovirus encoding either an Epac-S$^{H187}$ FRET-based sensor (*Klarenbeek et al., 2015*), or an AKAR3 FRET-based sensor (*Allen and Zhang, 2006*) addressed to the cytosol or the nucleus, at a multiplicity of infection of 1000 pfu/cell. Cells were cultivated for 36 hr prior to the experiments. Patch-clamp and IonOptix experiments were performed on cells 24 hr after dissociation.

## Confocal imaging

Ventricular cardiomyocytes from adult rats were plated on 35-mm glass bottom Petri dishes. Two hours after plating, the control cells were stained with commercial fluorescent $PEG_{5000}$ functionalized with FITC (Nanocs) or free Fluorescein (Sigma-Aldrich). Other cells were treated with neuraminidase (Sigma-Aldrich) for 1 hr at 37°C at a concentration of 0.25 U/ml, after which they were washed and re-incubated in physiological buffer solution. Cells were then stained with PEG-FITC. All molecules were dissolved in Ringer containing (in mM): NaCl 121.6, KCl 5.4, $MgCl_2$ 1.8, $CaCl_2$ 1.8, $NaHCO_3$ 4, $NaH_2PO_4$ 0.8, D-glucose 5, sodium pyruvate 5, HEPES 10, adjusted to pH 7.4. The acquisitions were made with a Leica TCS SP5 confocal microscope using a white light laser and an X40 oil immersion objective. The cells were excited at 495 nm and fluorescence was recovered at wavelengths >510 nm.

## Fluorescence intensity quantification

FITC or PEG-FITC that enters the T-tubules (TT) should flag the TTs, similar to Di4Anepps. Thus, we used the same established method used to measure the density of TT in cells. The center fluorescence image from each cell was selected for a thresholding analysis using an ImageJ plugin described previously (*Heinzel et al., 2002*) and TT quantification was performed using the ImageJ free software. On each image, a brightness/contrast correction was applied to correct acquisition settings differences. Images were filtered with a despeckle filter and a median filter. The image was then binarized using

an auto local threshold (Autothreshold plug-in). The threshold method and parameters were visually adjusted to ensure the coherence of the binarized image with the original one. A region of interest was defined inside the cell excluding the plasma membrane, and then a percent area was calculated. This value corresponds to the ratio of the number of white pixels over the size of the region multiplied by 100 and thus constitutes an index of the TT density where the fluorophore is visible.

## Detubulation

The TT network was chemically removed from a subset of control cells by a process termed detubulation as previously described (*Bourcier et al., 2019*). Briefly, cells were exposed to 300 µM imipramine for 15 min at room temperature, after which they were washed with physiological Ringer's solution and used for FRET. Detubulation was performed ≈1 hr before use in functional experiments. Detubulation was confirmed through confocal microscopy with 4 µM Di4Anepps. Cells were excited at 495 nm and fluorescence was recovered at wavelengths >510 nm.

## Molecular dynamics simulations

Molecular dynamics simulations of free Iso and PEG-Iso were carried out to predict their conformations in aqueous solution and to assess the solvent accessibility of Iso moiety once grafted to the PEG. A 3000-Da PEG molecule was used in these simulations. The 2D chemical structure of each of the two molecules was first converted into a three-dimensional structure using ChemAxon's Marvin-Sketch chemical editing software. The topology and GAFF force field parameters *Wang et al., 2004* used in this study were then automatically generated using the ACPYPE program (*Sousa da Silva and Vranken, 2012*). The initial three-dimensional structure was then placed in the center of a 13.4-nm side cubic simulation. These dimensions allow the solute not to interact with its virtually created images due to the periodic boundary conditions used to simulate an infinitely duplicated mesh. The box was then filled with water (model TIP3P; *Jorgensen et al., 1983*) and 150 mM NaCl. The system was then subjected to two short simulations (1 ns each) to equilibrate first the temperature around 300 K and then the pressure around 1 bar. The Nosé–Hoover (*Hoover, 1985*; *Nosé, 1984*) (coupling time $\tau T$ = 0.5 ps) and Parrinello–Rahman (*Parrinello and Rahman, 1981*) ($\tau P$ = 2.5 ps) algorithms were used for maintaining constant temperature and pressure, respectively. Finally, each of the two molecules was subjected to a production run of 150 ns. All simulations were performed using the GROMACS software version 2016 (*Abraham et al., 2015*). To analyze the trajectories, we used the GROMACS internal *gmx sasa* tool to calculate the SASA for Iso moiety in both free Iso or PEG-Iso molecules. To compare the solvent accessibility of the Iso moiety in free Iso and PEG-Iso, the percentage of PEG-Iso conformations with an $SASA_{PEG-Iso} \geq 0.9 \, SASA_{free-Iso}$ was calculated.

## Binding experiments

Binding assays were carried out in a final volume of 100 µl, containing a membrane suspension of CHO cells overexpressing human $\beta_1$- or $\beta_2$-adrenergic receptors, 145 pM of the radioligand [$^{125}$Iodo] cyanopindolol and non-radioactive ligands at concentrations ranging from 100 pM to 1 mM. Incubation was carried out for 2 hr at room temperature and terminated by addition of 4 ml PBS. Then, rapid filtration was performed through Whatman GF/C glass fiber filters previously soaked in PBS containing 0.33% polyethyleneimine using a compact cell harvester (Millipore 1225 Sampling Vacuum Manifold). The binding reaction was transferred to the filters and washed three times with Wash Buffer. Filter-bound radioactivity was measured in a gamma counter. All determinations were performed in triplicate at least.

## HTRF quantification of cAMP production

cAMP was quantified using the HTRF cAMP Gs dynamic detection kit (Revvity, Codolet, France) based on HTRF technology. The stimulation was done in the stimulation buffer (stimulation buffer 1X (Revvity) containing 1 mM 3-isobutyl-methylxanthine, IBMX). Briefly, HEK 293 cells stably expressing the human $\beta_1$- or $\beta_2$-ARs were resuspended in stimulation buffer and dispensed into 384-well plates (1000 cells per well (5 µl)) and stimulated with increasing concentrations (0.1 pM to 100 µM) of Iso diluted in stimulation buffer (5 µl/well) for 30 min at 20°C. Cells were then lysed, and cAMP levels were determined following manufacturer instructions. HTRF signal was read with the microplate reader Envision Excite (PerkinElmer, Turku, Finland).

## FRET imaging

FRET experiments were performed at room temperature 36 hr after cell plating. Cells were maintained in a Ringer solution containing (in mM): NaCl 121.6, KCl 5.4, $MgCl_2$ 1.8; $CaCl_2$ 1.8; $NaHCO_3$ 4, $NaH_2PO_4$ 0.8, D-glucose 5, sodium pyruvate 5, HEPES 10, adjusted to pH 7.4. Images were captured every 5 s using the 40x oil immersion objective of a Nikon TE 300 inverted microscope connected to a software-controlled (Metafluor, Molecular Devices, Sunnyvale, CA, USA) cooled charge coupled camera (Sensicam PE, PCO, Kelheim, Germany). Cells were excited for 150–300 ms by a Xenon lamp (100 W, Nikon, Champigny-sur-Marne, France) using a 440/20BP filter and a 455LP dichroic mirror. Dual emission imaging was performed using an Optosplit II emission splitter (Cairn Research, Faversham, UK) equipped with a 495LP dichroic mirror and BP filters 470/30 (CFP) and 535/30 (YFP), respectively. Spectral bleed-through into the YFP channel was subtracted using the formula: $YFP_{corr}$ = YFP − 0.6 × CFP.

## Electrophysiological experiments

The whole-cell configuration of the patch-clamp technique was used to record $I_{Ca,L}$. Patch electrodes had resistance of 0.5–1.5 MΩ when filled with internal solution containing (in mM): CsCl 118, EGTA 5, $MgCl_2$ 4, $Na_2$-phosphocreatine 5, $Na_2ATP$ 3.1, $Na_2GTP$ 0.42, $CaCl_2$ 0.062, HEPES 10, adjusted to pH 7.3 with CsOH. External $Cs^+$-Ringer solution contained (in mM): NaCl 107.1, CsCl 20, $NaHCO_3$ 4, $NaH_2PO_4$ 0.8, D-glucose 5, sodium pyruvate 5, HEPES 10, $MgCl_2$ 1.8, $CaCl_2$ 1.8, adjusted to pH 7.4 with NaOH. The cells were depolarized every 8 s from –50 to 0 mV for 400 ms. The liquid junction potential was adjusted to zero current between pipette and bath solution before the cells were attached to obtain a tight gigaseal (>1 GΩ). The use of –50 mV as holding potential allowed the inactivation of voltage-dependent sodium currents. Currents were not compensated for capacitance and leak currents. The amplitude of $I_{Ca,L}$ was measured as the difference between the peak inward current and the current at the end of the depolarization pulses. The cells were voltage-clamped with a patch-clamp amplifier (either RK-400 (Bio-Logic, Claix, France) or Axopatch 200B (Axon Instruments, Inc, Union City, CA, USA)). Currents were analog filtered at 5 kHz and digitally sampled at 10 kHz using a 16-bit analog to digital converter (DT321; Data Translation, Marlboro, MA, USA or a Digidata1440A, Axon Instruments) connected to a PC (Dell, Austin, TX, USA).

## Measurements of $Ca^{2+}$ transients and sarcomere shortening

All experiments were performed at room temperature. Isolated ARVMs were loaded with 1 µM Fura-2 AM (Thermo Fisher Scientific) and Pluronic acid (0.012%, Thermo Fisher Scientific) for 15 min in a Ringer solution containing (in mM): KCl 5.4, NaCl 121.6, sodium pyruvate 5, $NaHCO_3$ 4.013, $NaH_2PO_4$ 0.8, $CaCl_2$ 1.8, $MgCl_2$ 1.8, glucose 5, HEPES 10, pH 7.4 with NaOH. Sarcomere shortening and Fura-2 ratio (measured at 512 nm upon excitation at 340 and 380 nm) were simultaneously recorded in Ringer solution, using a double excitation spectrofluorimeter coupled with a video detection system (IonOptix, Milton, MA, USA). Myocytes were electrically stimulated, with biphasic field pulses (5 V, 4 ms) at a frequency of 1 Hz. $Ca^{2+}$ transient amplitude was measured by dividing the twitch amplitude (difference between the end-diastolic and the peak systolic ratios) by the end-diastolic ratio, thus corresponding to the percentage of variation in the Fura-2 ratio. Similarly, sarcomere shortening was assessed by its percentage variation, which was obtained by dividing the twitch amplitude (difference between the end-diastolic and the peak systolic sarcomere length) by the end-diastolic sarcomere length. Relaxation kinetics were estimated by a non-linear fit of the decaying part of the $Ca^{2+}$ transient and sarcomere shortening traces with the following equation: $X(t) = A \cdot \exp(-(t - t_0)/\tau) + A_0$, where $t_0$ is zero time, $A_0$ the asymptote of the exponential, $A$ the relative amplitude of the exponential, and $\tau$ the time constant of relaxation. The maximum first derivative of transients during the deflection allowed determination of the rising velocities of the signals. All parameters were calculated offline with dedicated software (IonWizard 6x, IonOptix R).

## Protein expression analysis

Cardiomyocytes were treated with or without isoprenaline 10 nM or PEG-Iso 1 µM (for 15 min) and the nuclei were enriched using a purification kit (ProteoExtract Subcellular Proteome, Millipore). Nuclear

proteins were examined by immunoblot for phosphorylation of PKA substrates using Phospho-(Ser/Thr) PKA Substrate antibody (Cell Signaling, Cat# 9621, RRID:AB_330304) and total protein per lane. The ratio of phospho-PKA expression was assessed by densiometric scanning of immunoblots.

## Statistical analysis

All results are expressed as mean ± SEM. Statistical analysis was performed using GraphPad software (GraphPad Software V.10.4.1, Inc, La Jolla, CA, USA). The normal distribution was tested by the Shapiro–Wilk normality test. For simple two-group comparison that satisfied parametric assumptions, we used an unpaired Student's $t$-test. Differences between multiple concentrations of the same drug were analyzed using an ordinary one-way ANOVA with Tukey post hoc test. A p-value < 0.05 was considered statistically significant.

## Acknowledgements

MB was supported by a doctoral grant from the Laboratory of Excellence LERMIT supported by the French National Research Agency (ANR-10-LABX-33) under the program 'Investissements d'Avenir' ANR-11-IDEX-0003-01. She also received a doctoral grant from the Fondation pour la Recherche Médicale. This work was also funded by grants ANR-15-CE14-0014-01 and ANR-23-CE14-0027-01 'CARDIOPEG' to RF which was used to support GM, alongside a postdoctoral research award grant from the Fondation Lefoulon-Delalande. Authors would like to thank K Leblanc (BioCIS) for help with preparative HPLC and Dr R Jockers (Institut Cochin, France), who kindly provided the HEK 293 cells expressing β-adrenergic receptors.

## Additional information

### Funding

| Funder | Grant reference number | Author |
|---|---|---|
| Agence Nationale de la Recherche | ANR-10-LABX-33 | Laurence Moine<br>Nicolas Tsapis<br>Rodolphe Fischmeister<br>Tâp Ha-Duong |
| Fondation pour la Recherche Médicale | | Marion Barthé |
| Agence Nationale de la Recherche | ANR-11-IDEX-0003-01 | Laurence Moine<br>Nicolas Tsapis<br>Rodolphe Fischmeister<br>Tâp Ha-Duong |
| Agence Nationale de la Recherche | ANR-15-CE14-0014-01 | Laurence Moine<br>Nicolas Tsapis<br>Rodolphe Fischmeister |
| Agence Nationale de la Recherche | ANR-23-CE14-0027-01 | Xavier Iturrioz<br>Tâp Ha-Duong |
| Fondation Lefoulon-Delalande | | George WP Madders |

The funders had no role in study design, data collection, and interpretation, or the decision to submit the work for publication.

### Author contributions

George WP Madders, Resources, Formal analysis, Investigation, Methodology, Writing – review and editing; Marion Barthé, Resources, Formal analysis, Investigation, Visualization, Methodology, Writing – original draft; Flora Lefebvre, Emilie Langlois, Florence Lefebvre, Resources, Formal analysis, Investigation; Patrick Lechêne, Catherine Llorens-Cortes, Investigation; Maya Dia, Formal analysis, Methodology; Xavier Iturrioz, Investigation, Methodology; Tâp Ha-Duong, Investigation, Methodology, Writing – original draft; Laurence Moine, Validation, Investigation, Methodology, Writing – original

draft; Nicolas Tsapis, Conceptualization, Supervision, Funding acquisition, Validation, Methodology, Writing – original draft, Project administration; Rodolphe Fischmeister, Conceptualization, Supervision, Funding acquisition, Validation, Methodology, Writing – original draft, Project administration, Writing – review and editing

### Author ORCIDs
George WP Madders (ID) https://orcid.org/0000-0003-1409-133X
Marion Barthé (ID) https://orcid.org/0000-0001-7631-9171
Flora Lefebvre (ID) https://orcid.org/0000-0003-3922-3001
Emilie Langlois (ID) https://orcid.org/0000-0002-3691-5849
Florence Lefebvre (ID) https://orcid.org/0000-0003-0133-5825
Patrick Lechêne (ID) https://orcid.org/0000-0002-9194-0000
Maya Dia (ID) https://orcid.org/0000-0001-6531-9176
Xavier Iturrioz (ID) https://orcid.org/0000-0001-7143-8323
Catherine Llorens-Cortes (ID) https://orcid.org/0000-0002-7667-0401
Tâp Ha-Duong (ID) https://orcid.org/0000-0002-0847-2948
Laurence Moine (ID) https://orcid.org/0000-0003-3306-9795
Nicolas Tsapis (ID) https://orcid.org/0000-0003-2095-2626
Rodolphe Fischmeister (ID) https://orcid.org/0000-0003-2086-9865

### Ethics
All animal care and experimental procedures complied with the ARRIVE guidelines and conform to the European Community guiding principles in the Care and Use of Animals (Directive 2010/63/EU of the European Parliament), the local Ethics Committee (CREEA Ile-de France Sud) guidelines, and the French decree no. 2013-118 of February 1, 2013 on the protection of animals used for scientific purposes (JORF no. 0032, February 7, 2013, p2199, text no. 24). Animal experiments according were carried out according to the European Community guiding principles in the care and use of animals (2010/63/UE), the local Ethics Committee (CREEA Ile-de-France Sud) guidelines and the French decree no. 2013-118 on the protection of animals used for scientific purposes.

### Decision letter and Author response
Decision letter https://doi.org/10.7554/eLife.84243.sa1
Author response https://doi.org/10.7554/eLife.84243.sa2

---

## Additional files

### Supplementary files
MDAR checklist

### Data availability
A dataset named CARDIOPEG has been uploaded to Zenodo. It contains all the individual experiments used for the figures in the article.

The following dataset was generated:

| Author(s) | Year | Dataset title | Dataset URL | Database and Identifier |
|---|---|---|---|---|
| Madders G, Fischmeister R | 2026 | Distinct functions of cardiac β-adrenergic receptors in the T-tubule vs outer surface membrane - Dataset | https://doi.org/10.5281/zenodo.19551791 | Zenodo, 10.5281/zenodo.19551791 |

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
