## [Editor Report]

This important study describes a novel approach using PEGylated isoprenaline to selectively activate β-adrenergic receptors in the surface sarcolemma relative to T-tubule sarcolemma of ventricular myocytes. Overall, the strength of evidence presented is convincing, and the authors present an interesting and impactful study that will be of interest to cardiac cell biologists and pharmacologists.

---

## [Decision Letter]

**Decision letter after peer review:**

Thank you for submitting your article "Distinct functions of cardiac β-adrenergic receptors in the T-tubule vs. outer surface membrane" for consideration by *eLife*. Your article has been reviewed by 3 peer reviewers, and the evaluation has been overseen by a Reviewing Editor and Richard Aldrich as the Senior Editor. The reviewers have opted to remain anonymous.

Essential Revisions:

1) Provide more rigorous and convincing evidence for the exclusion of PEG-FITC from the t-tubule (TT) network since this is foundational for the interpretation of the data. This can be provided by confocal Z-scan series comparing PEG-FITC and FITC in ARVM combined with quantification of T-tubule network density from multiple cells.

2) PEGylation increases the Ki for Iso for β2 adrenergic receptors (AR) by almost 1000-fold whereas the increase for β1 AR is about 200-fold. Thus, the relative contribution of β1 and β2 ARs to a 'comparable' dose of Iso and PEGylated Iso will potentially be different. This apparent difference complicates the interpretation of variances in effects between Iso vs PEG-Iso solely in terms of βAR localization to either OSM or TTs. Using antagonists to distinguish βAR subtype-specific responses would help address this.

3) A control experiment in which the effects of Iso and PEG-Iso are compared after neuraminidase treatment to see whether they have similar effects on nuclear PKA as might be predicted should be done.

4) Statistical analyses should be done comparing Iso vs PEG-Iso conditions rather than just different concentrations of the same drug.

*Reviewer #1 (Recommendations for the authors):*

Suggestions:

1) Is the differential expression of key cAMP/PKA signaling molecules (ACs, PKA, AKAPs, PDEs) critical in determining the response to ISO and PEG-ISO? If so, what are the mechanisms?

2) The work from the Dixon lab on CaV1.2 channel trafficking must be discussed and perhaps incorporated into the study. Could kinetic and amplitude differences in CaV1.2 channel insertion into the surface sarcolemma and TT membrane explain some of the findings of this study?

3) Perhaps mathematical modeling of cAMP/PKA dynamics would provide mechanistic insights.

4) Are there any physiological conditions in which surface and TT β-adrenergic receptors are differentially activated? Does the loss of TT during heart failure raise the signaling?

*Reviewer #2 (Recommendations for the authors):*

1) More robust evidence for lack of T-tubule penetration by PEGylated ligands with additional confocal images comparing groups and average data from multiple cells.

2) Demonstrate persistence of T-tubules in cultured ARVM infected with adenovirally delivered FRET-sensors.

3) Some progress on defining βAR subtype-specific responses. Just using antagonists for different subtypes would be helpful with PEG-Iso and Iso, because otherwise, data are not readily interpretable for this reviewer.

---

## [Author Response]

Essential Revisions:1) Provide more rigorous and convincing evidence for the exclusion of PEG-FITC from the t-tubule (TT) network since this is foundational for the interpretation of the data. This can be provided by confocal Z-scan series comparing PEG-FITC and FITC in ARVM combined with quantification of T-tubule network density from multiple cells.

As requested by the Reviewer, we have added additional data summarized in Figure 1G showing quantified fluorescence intensity for cells stained with FITC and PEG-FITC either in control (-) conditions or after treatment with neuraminidase. Statistical comparison between groups by one-way ANOVA supports the representative data and clearly demonstrates the inability of PEGylated ligands to enter the T-tubules, unless the glycocalyx has been cleaved.

2) PEGylation increases the Ki for Iso for β2 adrenergic receptors (AR) by almost 1000-fold whereas the increase for β1 AR is about 200-fold. Thus, the relative contribution of β1 and β2 ARs to a 'comparable' dose of Iso and PEGylated Iso will potentially be different. This apparent difference complicates the interpretation of variances in effects between Iso vs PEG-Iso solely in terms of βAR localization to either OSM or TTs. Using antagonists to distinguish βAR subtype-specific responses would help address this.

Please see lines 145-151. Functional experiments were performed using HEK-293 cells that stably express β_1_ or β_2_-ARs. Dose response curves to Iso and PEG-Iso showed that the relative differences between the drugs were of the same order of magnitude for β_1_ and β_2_-ARs. Thus, the ability of PEG-Iso to activate β_1_- and β_2_-ARs was reduced by the same order of magnitude for both receptors which should not bias the cellular experiments.

However as underlined by the reviewer, it is true and a very important point that functional differences in cAMP/PKA pathways may be due to differences in β_1_:β_2_ ratio on the T-tubule, compared to the ratio on the OSM. This could indeed contribute to the differences seen here and future studies will work on the development of pegylated β_1_- and β_2_-AR specific ligands to answer this question. Nevertheless, differences in β-AR subtypes in the TTM *vs.* OSM does not negate the conclusions in this paper where spatial differences are seen between spatially distinct β-ARs.

3) A control experiment in which the effects of Iso and PEG-Iso are compared after neuraminidase treatment to see whether they have similar effects on nuclear PKA as might be predicted should be done.

To answer this question, we proceeded in two ways. Firstly, we have used detubulation as a negative control. I.e. when T-tubules are removed the nuclear PKA response is the same for Iso as PEG-Iso at affinity adjusted concentrations (Figure 9 —figure supplement 4). Secondly, we have used neuraminidase as a positive control to demonstrate that when the glyocalyx is cleaved the nuclear cAMP/PKA response to PEG-ISO is increased through FRET experiments (Figure 9 —figure supplement 3).

4) Statistical analyses should be done comparing Iso vs PEG-Iso conditions rather than just different concentrations of the same drug.

We understand the point. But this cannot be easily done since the range of concentrations of Iso and PEG-Iso to produce a response on cytosolic and nuclear cAMP and PKA, or on I_Ca,L_, Ca^2+^ transients and contractility, differ by ~2 log units. So we have used a two-fold approach.

1. We have compared the effects of concentrations that differ by 100x magnitude (see lines 170-172, 212-213 and 217-219).

2. We have also sought a representation that would allow us to compare the effects of Iso and PEG-Iso independently of the concentrations used for each ligand. This is what led us to design Figure 9 —figure supplement 2 which compares the pattern of effects of Iso and PEG-Iso on cytosolic cAMP and PKA (Figure 9 —figure supplement 2A) or cytosolic cAMP and nuclear PKA (Figure 9 —figure supplement 2B). These two plots show clear differences between the effects of the two ligands. As written on Line 224-228, “It shows that for any given measured increase in [cAMP]_i_, PEG-Iso is more efficient than Iso to increase PKA activity in the cytosol (Figure 9 —figure supplement 2A), while on the contrary Iso is more efficient than PEG-Iso to increase PKA activity in the nucleus (Figure 9 —figure supplement 2B).” This, we believe, strongly supports the main conclusion of our study.

Reviewer #1 (Recommendations for the authors):Suggestions:1) Is the differential expression of key cAMP/PKA signaling molecules (ACs, PKA, AKAPs, PDEs) critical in determining the response to ISO and PEG-ISO? If so, what are the mechanisms?

PEGylation of Iso changes the pool of β-ARs that are activated. I.e. all the β-ARs (TT + OSM) for free Iso and only those on the OSM for PEG-Iso. Whilst the downstream cAMP/PKA signaling pathways may differ for the β-ARs depending on location, there is no reason to believe that this is due to the PEGylation of Iso itself compared to free Iso. Rather a β-AR in the same location should have the same response to Iso and PEG-Iso (at adjusted concentrations), thus activating the same pathways/signaling molecules. Observed differences are due to the spatial differences of the β-ARs.

2) The work from the Dixon lab on CaV1.2 channel trafficking must be discussed and perhaps incorporated into the study. Could kinetic and amplitude differences in CaV1.2 channel insertion into the surface sarcolemma and TT membrane explain some of the findings of this study?

Lines 377-380. This has been incorporated in so much as the same change was seen from total β-AR stimulation to those in only the OSM. To truly elucidate changes in Ca_V_1.2 trafficking and differential kinetics based upon location using our approach it would be necessary to block the different Ca_V_1.2 populations, e.g. using PEG-Nicardipine. This is beyond the scope of our study but raises the potential applications that this technology has.

3) Perhaps mathematical modeling of cAMP/PKA dynamics would provide mechanistic insights.

See answer to point 1. Our main mechanistic insight is the differential regulation of cytoplasmic and nuclear cAMP/PKA dependent on spatial β-AR activation. A further study is required to fully elucidate the role of different PDEs etc. in regulating these pathways.

4) Are there any physiological conditions in which surface and TT β-adrenergic receptors are differentially activated? Does the loss of TT during heart failure raise the signaling?

A very interesting point. Differences in spatial expression of β-ARs are thought to occur alongside the loss of T-tubules in heart failure. Whilst here our focus has been on the proof of principle of spatial restriction, we would very much hope that future studies could apply this technology in pathologies such as heart failure.

Reviewer #2 (Recommendations for the authors):1) More robust evidence for lack of T-tubule penetration by PEGylated ligands with additional confocal images comparing groups and average data from multiple cells.

Please see our response to ‘essential revision #1’. Figure 1G now provides summary data, quantifying fluorescence intensity for cells stained with FITC and PEG-FITC either in control (-) conditions or after treatment with neuraminidase. Statistical comparison between groups by one-way ANOVA supports the representative data and clearly demonstrates the inability of PEGylated ligands to enter the T-tubules, unless the glycocalyx has been cleaved.

2) Demonstrate persistence of T-tubules in cultured ARVM infected with adenovirally delivered FRET-sensors.

It is necessary to wait ≈36 hours for the adenovirus to infect and stably express the compartmentalized cAMP or PKA sensors. We have now quantified the T-tubule network over this experimental time frame in Figure 9 —figure supplement 6 and show that whilst the T-tubule density is indeed decreased by our maximal time point of 48 hours there is in fact still a robust T-tubule network within the cell. This decrease in T-tubules actually suggests that if it were possible to perform FRET immediately on freshly isolated ARVMs we would see a much larger discrepancy between Iso and PEG-Iso where differences were observed.

3) Some progress on defining βAR subtype-specific responses. Just using antagonists for different subtypes would be helpful with PEG-Iso and Iso, because otherwise, data are not readily interpretable for this reviewer.

Please see our response to ‘essential revision #2’ and Lines 145-151. Functional experiments were performed using HEK-293 cells that stably express either the β_1_- or β_2_-ARs. Dose response curves to Iso and PEG-Iso showed that the relative differences between the drugs were of the same order of magnitude for β_1_ and β_2_-ARs. Thus, the ability of PEG-Iso to activate β_1_- and β_2_-ARs was reduced by the same order of magnitude for both receptors which should not bias the cellular experiments.

However as underlined by the reviewer, it is true and a very important point that functional differences in cAMP/PKA pathways may be due to differences in β_1_:β_2_ ratio on the T-tubule, compared to the ratio on the OSM. This could indeed contribute to the differences seen here and future studies will work on the development of pegylated β_1_ and β_2_ specific ligands to answer this question. However, differences in β-AR subtype on the T-tubule *vs.* the OSM does not negate the conclusions in this paper where spatial differences are seen between spatially distinct β-ARs.